# Normal-Abnormal Guided Generalist Anomaly Detection

**Yuexin Wang**[1,2*]**, Xiaolei Wang**[1,2*]**, Yizheng Gong**[1,2]**, Jimin Xiao**[1†]
[1]Xi'an Jiaotong-Liverpool University
[2]University of Liverpool

## Abstract

Generalist Anomaly Detection (GAD) aims to train a unified model on an original domain that can detect anomalies in new target domains. Previous GAD methods primarily use only normal samples as references, overlooking the valuable information contained in anomalous samples that are often available in real-world scenarios. To address this limitation, we propose a more practical approach: normal-abnormal-guided generalist anomaly detection, which leverages both normal and anomalous samples as references to guide anomaly detection across diverse domains. We introduce the Normal-Abnormal Generalist Learning (NAGL) framework, consisting of two key components: Residual Mining (RM) and Anomaly Feature Learning (AFL). RM extracts abnormal patterns from normal-abnormal reference residuals to establish transferable anomaly representations, while AFL adaptively learns anomaly features in query images through residual mapping to identify instance-aware anomalies. Our approach effectively utilizes both normal and anomalous references for more accurate and efficient cross-domain anomaly detection. Extensive experiments across multiple benchmarks demonstrate that our method significantly outperforms existing GAD approaches. This work represents the first to adopt a mixture of normal and abnormal samples as references in generalist anomaly detection. The code and datasets are available at `https://github.com/JasonKyng/NAGL`.

## 1 Introduction

Visual Anomaly Detection (AD) [13, 30, 63, 5, 14, 67, 56, 59, 18, 17, 41, 6] plays a crucial role in industrial quality inspection [2, 48, 69, 25] and medical diagnosis [14, 20]. Its primary objectives are to classify images as normal or anomalous and to localize anomalies within those images. Traditional AD methods [31, 8, 15, 53, 9] focus on training and testing a model on a single domain, without considering how detection capabilities might transfer to a different target domain. However, many real-world AD scenarios prohibit training on the target domain due to data scarcity and privacy issues, making it difficult to achieve the desired outcome in that target domain. To address the challenge, InCTRL [68] and ResAD [60] propose Generalist Anomaly Detection (GAD) that aims to train a unified model on the original domain while enabling AD on the target domain. As shown in Fig. 1a, the GAD framework adopts a meta-learning strategy. This strategy trains the model in the original domain to localize the anomalous regions of a query image by referring to a limited number of normal references. Subsequently, the learned ability is transferred to the new target domain.

Although previous GAD methods [68, 60] have made substantial progress, they are not yet practical for real-world applications. Models trained exclusively on normal samples often lack the discriminative power to reliably distinguish anomalies [14, 61]. However, real-world scenarios often provide a small number of anomalous samples (*e.g.*, defective parts or diagnosed disease cases). These anomalous

---

[*]Equal Contribution.
[†]Corresponding Author.

39th Conference on Neural Information Processing Systems (NeurIPS 2025).

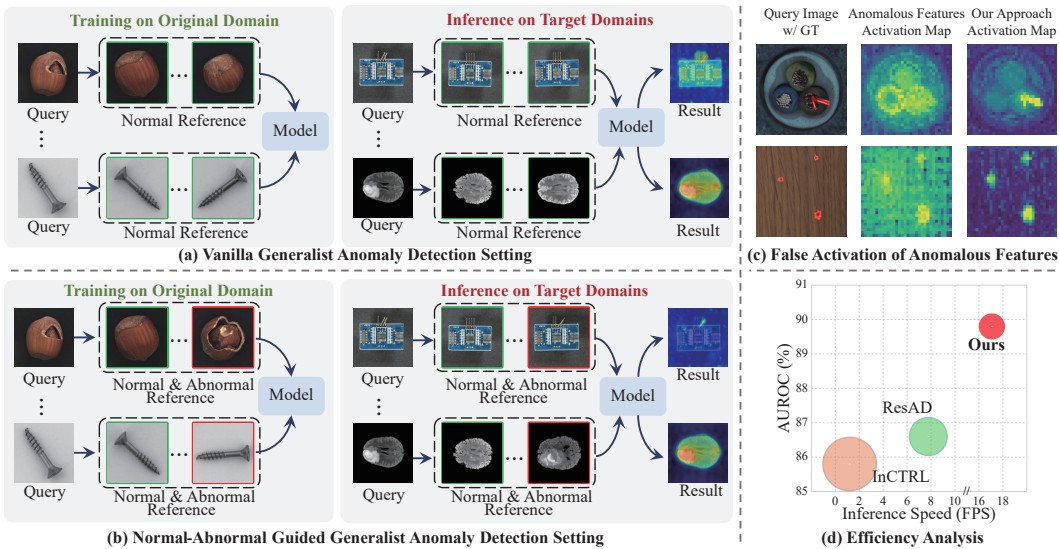

Figure 1: Overview of existing and proposed GAD paradigms. (a) Vanilla GAD only adopts a few normal samples as references. (b) Our approach combines normal and abnormal references to enhance detection. (c) Direct application of the KNN-based method to our normal-abnormal guided detection task causes to false activations (middle), while our approach eliminates them (right). (d) Comparisons of different methods in terms of AUROC sample (y-axis), inference speed (x-axis), and model size (circle radius). Our approach achieves the **highest** AUROC sample metric for anomaly detection while being **2× faster** than ResAD and **14× faster** than InCTRL.

samples contain valuable information on anomaly characteristics that could be leveraged to improve detection. Given this context, we propose a more practical and effective approach: normal-abnormal-guided generalist anomaly detection (illustrated in Fig. 1b). This approach leverages both normal and anomalous samples as references, guiding the model to detect anomalies across diverse domains. The core of this approach is learning the relationships between a query and these references in the original domain, and then applying this learned understanding to the target domain.

Previous methods [68, 60] leverage residuals between queries and normal references to ensure transferability, but these approaches are specifically tailored for scenarios where few-shot normal samples are provided, which cannot be adapted to the normal-abnormal guided paradigm. When provided with a reference set with both normal and abnormal samples, KNN-based approaches [48, 12] can serve as a solution, where sample regions far away from normal references and close to abnormal references are considered as abnormal. However, these KNN-based approaches are typically training-free, lacking the adaptability offered by data-driven learning. Additionally, we experimentally find that the KNN-based method suffers from false activation problems (see Fig. 1c). To address these limitations, we propose a Normal-Abnormal Generalist Learning (NAGL) framework, which utilizes the residual features to model the differences between normal and anomalous references. The proposed method achieves superior detection performance at faster speeds (see Fig. 1d), demonstrating its significant potential for practical applications.

The core of our NAGL framework consists of two parts: Residual Mining (RM) from normal-abnormal references and Anomaly Feature Learning (AFL) for query images through residual mapping. To fully explore abnormal reference patterns and maintain transferability, RM leverages normal-abnormal residuals to learn abnormal reference patterns through a designed attention operation, obtaining residual proxies. AFL adaptively learns abnormal features of a query image by comparing residual proxies with residuals between the query feature and normal references, obtaining anomaly proxies. Finally, anomaly localization results are acquired by similarity computation between the query feature and anomaly proxies. Our NAGL only relies on residual training to adaptively capture the similarities and differences between query and normal-abnormal reference samples, so it can be transferred between different domains.

Leveraging RM and AFL, our proposed NAGL framework effectively achieves normal-abnormal guided generalist anomaly detection. The main contributions of this work are summarized as follows:

- We propose a different generalist anomaly detection task and a corresponding dataset split. This task is the first to adopt a mixture of normal and abnormal samples as references.

- We propose a novel Normal-Abnormal Generalist Learning framework to effectively leverage abnormal reference in GAD that adapts normal-abnormal reference residuals to mine potential anomalies in the query.

- Extensive experiments across multiple anomaly detection benchmarks demonstrate that our method significantly outperforms existing GAD approaches.

## 2 Related Work

### 2.1 Anomaly Detection

Artificial intelligence techniques based on deep learning have been widely applied [27, 38, 57], with anomaly detection (AD) being a significant application. AD can be divided into various tasks according to real-world requirements, *e.g.*, unsupervised AD [55, 54, 69, 37, 64, 49], few-shot AD (FSAD) [21, 15, 28, 42, 50, 33], zero-shot AD (ZSAD) [23, 65, 7], noisy AD [8, 26], 3D AD [16, 11, 36, 66], and open-set AD [14, 67]. Among these tasks, unsupervised AD generally adopts a one-class classification paradigm to train the detection model, *i.e.*, the model is only trained on normal samples and can detect unseen abnormal patterns during the inference phase. Existing unsupervised methods can be divided into three main categories: reconstruction-based [51, 40, 19], feature-embedding-based [44, 48, 29, 58], and augmentation-based [62, 64, 34] methods. PatchCore [48], a simple feature-embedding-based method, firstly constructs a memory bank of normal embedding features, then introduces a nearest neighbour search to find several nearest neighbours for each test embedding feature, and computes the distance between the test embedding feature and its neighbours as an anomaly score. PatchCore can also be extended to a few-shot AD task well, where the used memory bank only requires a few normal reference samples to construct. Based on this work [48], FastRecon [15] leverages ridge regression on normal features to quickly reconstruct test features. However, these methods only have a testing phase and detect anomalies on a single domain of data, lacking transferability. Therefore, some ZSAD [65] and GAD [60, 68] methods have begun to study cross-domain detection. And [43] also proposes a general few-shot defect classification framework that addresses real-world applications of defect type identification. Building upon the GAD task, our proposed normal-abnormal guided generalist AD framework addresses cross-domain anomaly detection by effectively utilizing abnormal samples from the original domain to improve performance.

### 2.2 Generalist Anomaly Detection

FSAD is designed to identify anomalies using only a limited number of normal samples from target datasets. Existing FSAD methods can be divided into training-free-based [48, 15] and meta-learning-based methods [21, 68, 60]. Meta-learning-based methods focus on generalizing the detection of the model to the new target domain. RegAD [21] trains a registration network using samples from seen categories, which can register samples from unseen categories and achieve cross-category anomaly detection. However, RegAD remains dependent on domain relevance between training and testing data. To achieve domain-agnostic anomaly detection, InCTRL [68] firstly proposes a generalist anomaly detection (GAD) task and utilizes residual distance to discriminate anomalies. Subsequently, ResAD [60] applies residual features to eliminate domain dependencies to implement the GAD task. The reference set used by the vanilla GAD task only contains normal samples. However, in practical scenarios, a small number of abnormal samples can also be obtained. Therefore, we propose the normal-abnormal guided GAD task, where the reference set combines normal and abnormal samples.

## 3 Methodology

### 3.1 Normal-Abnormal Guided GAD Task

In the normal-abnormal guided GAD task, we focus on training a unified model on the original domain and detecting anomalies in a new target domain using both normal and abnormal samples

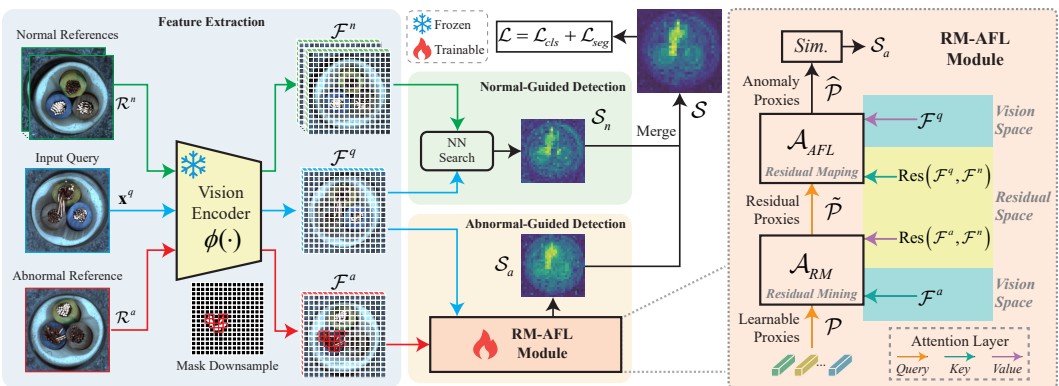

Figure 2: Overview of our proposed NAGL framework. Given a test image and its corresponding reference images (normal and abnormal), features are extracted through a pre-trained backbone network. The extracted normal features guide the generation of a normal-guided score map. Meanwhile, abnormal features are processed through the RM-AFL module to produce an abnormal-guided score map. This module implements a transformation process: from learnable proxies in vision space to residual proxies capturing normal-abnormal differences in residual space, and finally to anomaly proxies in vision space that highlight specific anomalous regions in the query image. The final anomaly score is computed by merging both normal and abnormal guided score maps.

as references. Formally, let $\mathcal{D}_{origin} = \{\mathcal{X}, \mathcal{Y}\}$ be the training dataset from original domain, where $\mathcal{X} = \{\mathbf{x}_i\}_{i=1}^N$ consists of $N$ normal and abnormal samples, $\mathcal{Y} = \{\mathbf{y}_i\}_{i=1}^N$ is corresponding ground truth masks, and each normal sample is equipped a zero mask. During training phase, we organize the data into many episodes, where each episode consists of a reference set $\mathbf{R} = \{\mathcal{R}^n, \mathcal{R}^a\}$ and a query input $\mathbf{x}^q$ from $\mathcal{D}_{origin}$. The reference set $\mathbf{R}$ contains normal samples $\mathcal{R}^n = \{r_k^n\}_{k=1}^{K_1}$ and abnormal samples $\mathcal{R}^a = \{r_k^a\}_{k=1}^{K_2}$, $K_1$ and $K_2$ denote the number of normal and abnormal reference samples, respectively. We train a unified detection model on $\mathcal{D}_{origin}$. During inference, the model is evaluated on the target domain dataset $\mathcal{D}_{target}$ containing unseen categories compared to $\mathcal{D}_{origin}$. The testing process is consistent with the training phase, where each test sample is equipped with a normal-abnormal reference set. In our proposed normal-abnormal guided GAD task, considering the scarcity of abnormal samples, we set $K_1 \geq K_2$, making it highly applicable to real-world situations.

### 3.2 Overview of Proposed Method

Our proposed Normal-Abnormal Generalist Learning (NAGL) framework is designed to detect anomalies in the target domain through original domain training. As shown in Fig. 2, given a query input and normal-abnormal references, we extract these features by applying a pre-trained backbone network for further processing. After feature extraction, we directly leverage the Nearest Neighbor (NN) search between the query and normal references, obtaining an initial anomaly score map. Next, to apply an abnormal reference to improve the initial anomaly map, we propose an RM-AFL module. This module consists of two attention parts: Residual Mining (RM) from normal-abnormal references and Anomaly Feature Learning (AFL) for query images with residual mapping. RM relies on normal-abnormal residuals to learn the representation for abnormal reference patterns, generating several residual proxies. AFL applies these proxies to learn abnormal areas in the query image and obtain anomaly proxies. Finally, we compute the cosine similarity between the query feature and anomaly proxies for anomaly localization results.

### 3.3 Feature Extraction

In this section, we describe the feature extraction process for an episode. For each episode, we have a normal-abnormal reference set $\mathbf{R} = \{\mathcal{R}^n, \mathcal{R}^a\}$ and a input query image $\mathbf{x}^q \in \mathbb{R}^{H \times W \times 3}$. For input query and references, we follow the common practice of using a pre-trained backbone network $\phi(\cdot)$ to extract their features. Subsequently, we obtain query patch features $\mathcal{F}^q = \{f_i^q\}_{i=1}^L$, normal reference patch features $\mathcal{F}^n = \{f_i^n\}_{i=1}^{K_1 L}$, and abnormal reference patch features $\mathcal{F}^a = \{f_i^a\}_{i=1}^{K_2 L}$, where

$f_i^q, f_i^n, f_i^a \in \mathbb{R}^C$, $L = h * w$ is the number of patches in one feature map, $h$ and $w$ are length and width of feature map, and $C$ is the channel dimension of each patch feature. For abnormal reference features, we downsample the anomaly mask to match the length of $\mathcal{F}^a$, yielding $\mathcal{M}^a \in \mathbb{R}^{K_2 L}$.

### 3.4 Normal-Guided Anomaly Score

In this section, we apply query and normal reference features to obtain an initial anomaly localization result. Following the previous work [48], we directly apply the nearest neighbor search between query features $\mathcal{F}^q$ and normal reference features $\mathcal{F}^n$ to generate an anomaly score map $\mathcal{S}_n \in \mathbb{R}^L$. Specifically, we apply the NN method to find the closest reference patch feature $f_*^n$ for each query patch feature $f_i^q$ in normal reference patches $\mathcal{F}^n$,

$$f_*^n := \underset{f^n \in \mathcal{F}^n}{\operatorname{argmin}} \left( \mathbf{d}(f_i^q, f^n) \right), \tag{1}$$

where $\mathbf{d}(\cdot, \cdot)$ is the cosine distance function and $i \in \{1, 2, \cdots, L\}$. Therefore, each patch anomaly score is defined as

$$\mathcal{S}_n^i = \mathbf{d}(f_i^q, f_*^n), \tag{2}$$

where $\mathcal{S}_n^i = 0$ indicates the most normal region and $\mathcal{S}_n^i = 1$ indicates the most abnormal region.

### 3.5 Abnormal-Guided Anomaly Score

In this section, we focus on applying abnormal reference to improve the initial anomaly map. This section mainly consists of Residual Mining (RM) from normal-abnormal reference and Anomaly Feature Learning (AFL) for query images by residual mapping. The whole RM-AFL mainly relies on residual training to adaptively capture the similarities and differences between query and reference samples, so it can be transferred between different domains.

#### 3.5.1 Residual Mining from References

The RM module introduces a learnable query proxy to adaptively capture abnormal patterns in anomalous references and utilizes normal-abnormal residuals to learn anomalous variations through a cross-attention mechanism. Specifically, for given abnormal reference features $\mathcal{F}^a$, we employ the NN method to identify the closest normal reference patch feature for each $f_i^a$ in $\mathcal{F}^a$ and compute the residual between the two. For simplicity, we denote this process as

$$\operatorname{Res}(\mathcal{F}^a, \mathcal{F}^n) = \mathcal{F}^a - \mathcal{F}_*^n, \tag{3}$$

where $\mathcal{F}_*^n \in \mathbb{R}^{K_2 L \times C}$ consists of $f_*^n$ corresponding to each $f_i^a$. We then design an attention layer $(\mathcal{A}_{RM})$ that utilizes normal-abnormal residuals to learn representations of abnormal variations, thereby generating residual proxies. We state *Query*, *Key*, and *Value* in the attention computation:

$$\mathbf{Q}_1 = \mathbf{W}_1^{\mathcal{Q}} \mathcal{P}, \mathbf{K}_1 = \mathbf{W}_1^{\mathcal{K}} \mathcal{F}^a, \mathbf{V}_1 = \mathbf{W}_1^{\mathcal{V}} \operatorname{Res}(\mathcal{F}^a, \mathcal{F}^n), \tag{4}$$

where $\mathcal{P} \in \mathbb{R}^{M \times C}$ is learnable proxies and randomly initialized, parameters $\mathbf{W}_1^{\mathcal{Q}}$, $\mathbf{W}_1^{\mathcal{K}}$, and $\mathbf{W}_1^{\mathcal{V}}$ transform input features into query, key, and value vectors, $M$ is a hyperparameter representing the number of $\mathcal{P}$. Therefore, the attention-based RM mechanism can be described as

$$\widetilde{\mathcal{P}} = \mathbf{SA}_1 \left( \operatorname{Softmax} \left( \frac{\mathbf{Q}_1 \mathbf{K}_1^{\mathrm{T}}}{\sqrt{d}} + \mathcal{M}' \right) \mathbf{V}_1 \right), \tag{5}$$

where $\widetilde{\mathcal{P}} \in \mathbb{R}^{M \times C}$ is called residual proxies, $\mathbf{SA}_1(\cdot)$ denotes self-attention operation, and $d$ is a scaling factor [52]. The attention mask $\mathcal{M}' = \alpha(1 - \mathcal{M}^a)$ contains either zero or negative infinity values, where $\alpha$ is a large negative value (*e.g.*, $-10^9$). This mask ensures cross-attention focuses exclusively on abnormal regions. Residual proxies $\widetilde{\mathcal{P}}$ learn abnormal reference patterns and apply residuals as *Value* to represent variations for anomalies.

#### 3.5.2 Anomaly Feature Learning for Query Images

In this section, AFL focuses on applying obtained residual proxies $\widetilde{\mathcal{P}}$ to learn potential abnormal patterns in $\mathcal{F}^q$, obtaining anomaly proxies. The anomaly localization results are acquired by similarity

computation between query patch features and anomaly proxies. This process can also be described by another attention layer ($\mathcal{A}_{AFL}$). The *Query*, *Key*, *Value* of the attention module is designed as:

$$\mathbf{Q}_2 = \mathbf{W}_2^{\mathcal{Q}}\widetilde{\mathcal{P}}, \mathbf{K}_2 = \mathbf{W}_2^{\mathcal{K}}\mathrm{Res}(\mathcal{F}^q, \mathcal{F}^n), \mathbf{V}_2 = \mathbf{W}_2^{\mathcal{V}}\mathcal{F}^q, \tag{6}$$

where $\mathbf{W}_2^{\mathcal{Q}}$, $\mathbf{W}_2^{\mathcal{K}}$, $\mathbf{W}_2^{\mathcal{V}}$ are learnable parameters. Our proposed AFL can be written as:

$$\widehat{\mathcal{P}} = \mathbf{SA}_2\left(\mathrm{Softmax}\left(\frac{\mathbf{Q}_2\mathbf{K}_2^{\mathrm{T}}}{\sqrt{d}}\right)\mathbf{V}_2\right), \tag{7}$$

where $\mathbf{SA}_2(\cdot)$ denotes another self-attention module different from $\mathbf{SA}_1(\cdot)$, $\widehat{\mathcal{P}} \in \mathbb{R}^{M \times C}$ is called anomaly proxies. $\widehat{\mathcal{P}}$ captures abnormal patterns in $\mathcal{F}^q$ by comparing reference residuals $\mathrm{Res}(\mathcal{F}^a, \mathcal{F}^n)$ with the query-normal residuals $\mathrm{Res}(\mathcal{F}^q, \mathcal{F}^n)$. $\widehat{\mathcal{P}}$ has the most distinguishable patch feature information in $\mathcal{F}^q$. Therefore, we calculate the mean of similarity between each anomaly proxy $\widehat{\mathcal{P}}_m$ and the query patch features $\mathcal{F}^q$ as an anomaly-guided anomaly score map $\mathcal{S}_a$. Each patch anomaly score is

$$\mathcal{S}_a^i = \frac{1}{M}\sum_{m=1}^{M} 1 - \mathbf{d}(f_i^q, \widehat{\mathcal{P}}_m), \tag{8}$$

where $\mathcal{S}_a \in \mathbb{R}^L$. Finally, we merge the initial normal-guided score map and abnormal-guided score map to acquire final anomaly localization results:

$$\mathcal{S} = \mathcal{S}_n + \mathcal{S}_a. \tag{9}$$

Following [12], we calculate the image-level anomaly score by averaging the top $1\%$ highest values in the score map $\mathcal{S} \in \mathbb{R}^L$. Specifically, we denote this operation as $s = \mathcal{T}_{0.01}(\mathcal{S})$, where $\mathcal{T}_{0.01}(\mathcal{S})$ represents the average of the $1\%$ highest scores in $\mathcal{S}$.

### 3.6 Training on Original Domain

The whole NAGL framework is trained on original domain data. Our optimization objective is that the predicted map $\mathcal{S}$ should be consistent with the query ground truth mask $\mathcal{M}^q$. Therefore, we apply Focal loss [35] and Dice loss [32] to achieve anomaly segmentation training, *i.e.*,

$$\mathcal{L}_{\mathrm{seg}} = \mathbf{Focal}(\mathcal{S}, \mathcal{M}^q) + \mathbf{Dice}(\mathcal{S}, \mathcal{M}^q), \tag{10}$$

where $\mathcal{M}^q$ denotes the downsampled and reshaped ground truth mask of the query image, $\mathbf{Focal}(\cdot)$ and $\mathbf{Dice}(\cdot)$ represent the Focal loss and Dice loss, respectively. Moreover, we also guarantee that the image-level predicted label is consistent with the classification label of the query image. Binary Cross-Entropy (BCE) loss is leveraged to optimize anomaly classification,

$$\mathcal{L}_{\mathrm{cls}} = \mathbf{BCE}(s, y^q), \tag{11}$$

where $s$ denotes predicted image-level anomaly score, $y^q$ is ground truth classification label, and $\mathbf{BCE}(\cdot)$ denotes BCE loss function. Finally, a hyper-parameter $\lambda$ is used to balance the classification and segmentation losses,

$$\mathcal{L} = \mathcal{L}_{\mathrm{cls}} + \lambda\mathcal{L}_{\mathrm{seg}}. \tag{12}$$

### 3.7 Inference on Target Domain

During inference, for given target domain data $D_{target}$, similar to the training phase, we first use the NN method to obtain a normal-guided score map $\mathcal{S}_n$. Next, we apply RM and AFL modules to generate an abnormal-guided score map $\mathcal{S}_a$. According to Eq. (9), we merge the two score maps as the final score map $\mathcal{S} \in \mathbb{R}^L$. Next, the size of $\mathcal{S}$ is reshaped to $\mathbb{R}^{h \times w}$, and up-sampled to $\mathbb{R}^{H \times W}$.

## 4 Experiment

### 4.1 Experimental Setup

**Datasets.** To validate the efficiency of our NAGL framework, we construct three benchmarks using the MVTecAD [2], VisA [69], and BraTS [45] datasets. The benchmarks include (1) training on

Table 1: Comparison of the proposed method with the previous methods on MVTecAD and VisA datasets. $N^i$ and $A^i$ represent the number of normal and abnormal reference samples in $i$-shot learning, respectively. Results marked with † are quoted from [23], while those marked with ∗ are based on our re-implementation. The best/runner-up results are highlighted in **bold**/underline.

| Setting | Method | MVTecAD | | | | | | VisA | | | | | |
| | | Image-level | | | Pixel-level | | | Image-level | | | Pixel-level | | |
| | | AUROC | AP | F1-max | AUROC | PRO | F1-max | AUROC | AP | F1-max | AUROC | PRO | F1-max |
| $N^1$ | SPADE† [10] | 81.0 | 90.6 | 90.3 | 91.2 | 83.9 | 42.4 | 79.5 | 82.0 | 80.7 | 95.6 | 84.1 | 35.5 |
| | PaDiM† [13] | 76.6 | 88.1 | 88.2 | 89.3 | 73.3 | 40.2 | 88.2 | 62.8 | 75.3 | 89.9 | 64.3 | 17.4 |
| | PatchCore† [48] | 83.4 | 92.2 | 90.5 | 92.0 | 79.7 | 50.4 | 79.9 | 82.8 | 81.7 | 95.4 | 80.5 | 38.0 |
| | WinCLIP† [23] | 93.1 | 96.5 | 93.7 | 95.2 | 87.1 | 55.9 | 83.8 | 85.1 | 83.1 | 96.4 | 85.1 | 41.3 |
| | PromptAD [31] | 94.6 | 97.1 | - | 95.9 | 87.9 | - | 86.9 | 88.4 | - | 96.7 | 85.1 | - |
| | ResAD∗ [60] | 84.8 | 92.7 | 91.2 | 93.4 | 83.3 | 48.2 | 80.9 | 83.7 | 81.3 | 95.9 | 79.6 | 37.8 |
| $N^1 + A^1$ | **Ours** | **95.8** | **97.5** | **95.7** | **96.6** | **92.9** | **58.9** | **88.5** | **89.4** | **85.4** | **97.5** | **91.1** | **41.8** |
| $N^2$ | SPADE† [10] | 82.9 | 91.7 | 91.1 | 92.0 | 85.7 | 44.5 | 80.7 | 82.3 | 81.7 | 96.2 | 85.7 | 40.5 |
| | PaDiM† [13] | 78.9 | 89.3 | 89.2 | 91.3 | 78.2 | 43.7 | 67.4 | 71.6 | 75.7 | 92.0 | 70.1 | 21.1 |
| | PatchCore† [48] | 86.3 | 93.8 | 92.0 | 93.3 | 82.3 | 53.0 | 81.6 | 84.8 | 82.5 | 96.1 | 82.6 | 41.0 |
| | WinCLIP† [23] | 94.4 | 97.0 | 94.4 | 96.0 | 88.4 | 58.4 | 84.6 | 85.8 | 83.0 | 96.8 | 86.2 | **43.5** |
| | PromptAD [31] | 95.7 | 97.9 | - | 96.2 | 88.5 | - | 88.3 | 90.0 | - | 97.1 | 85.8 | - |
| | InCTRL [68] | 94.0 | 96.9 | - | - | - | - | 85.8 | 87.7 | - | - | - | - |
| | ResAD∗ [60] | 87.2 | 93.9 | 92.2 | 94.8 | 85.5 | 50.2 | 86.6 | 88.3 | 84.1 | 96.5 | 82.3 | 40.0 |
| $N^2 + A^1$ | **Ours** | **96.8** | **97.9** | **96.3** | **96.8** | **93.2** | **59.8** | **89.8** | **90.6** | **86.8** | **97.6** | **91.4** | 43.3 |
| $N^4$ | SPADE† [10] | 84.8 | 92.5 | 91.5 | 92.7 | 87.0 | 46.2 | 81.7 | 83.4 | 82.1 | 96.6 | 87.3 | 43.6 |
| | PaDiM† [13] | 80.4 | 90.5 | 90.2 | 92.6 | 81.3 | 46.1 | 72.8 | 75.6 | 78.0 | 93.2 | 72.6 | 24.6 |
| | PatchCore† [48] | 88.8 | 94.5 | 92.6 | 94.3 | 84.3 | 55.0 | 85.3 | 87.5 | 84.3 | 96.8 | 84.9 | 43.9 |
| | WinCLIP† [23] | 95.2 | 97.3 | 94.7 | 96.2 | 89.0 | 59.5 | 87.3 | 88.8 | 84.2 | 97.2 | 87.6 | **47.0** |
| | PromptAD [31] | 96.6 | 98.5 | - | 96.5 | 90.5 | - | 89.1 | 90.8 | - | 97.4 | 86.2 | - |
| | InCTRL [68] | 94.5 | 97.2 | - | - | - | - | 87.7 | 90.2 | - | - | - | - |
| | ResAD∗ [60] | 90.7 | 95.7 | 93.9 | 95.8 | 88.7 | 53.0 | 89.3 | 90.7 | 86.5 | 96.8 | 84.1 | 41.6 |
| $N^4 + A^1$ | **Ours** | **97.1** | **98.0** | **96.4** | **97.0** | **93.5** | **60.1** | **91.2** | **91.7** | **87.8** | **97.8** | **91.5** | 44.0 |

VisA and testing on MVTecAD, (2) training on MVTecAD and testing on VisA, and (3) training on MVTecAD and testing on BraTS. The first two benchmarks assess generalization capabilities across different industrial domains, while the third evaluates cross-domain transfer capabilities from industrial to medical applications. During inference, the normal references are sampled from the training set of target datasets, while abnormal references are sampled from each anomaly type. Further details are provided in Appendix A.

**Evaluation Metrics.** We evaluate both image-level anomaly classification and pixel-level segmentation performance using three metrics for each task, following previous works [23, 12, 60]. For detection performance, we employ the Area Under the Receiver-Operator Curve (AUROC, [13]), the maximum F1-score at the optimal threshold (F1-max), and the Average Precision (AP), calculated using image-level anomaly scores. Similarly, for segmentation performance, we utilize the AUROC, F1-max, and per-region overlap (PRO) [3, 2] metrics, computed using pixel-wise anomaly scores.

**Implementation Details.** Our approach employs the ViT-based [47] model as the vision encoder, specifically utilizing its most lightweight version (ViT-S, $21M$ parameters) to ensure low latency in practical applications. We freeze the parameters of the vision encoder throughout the experiments and only update the parameters of the proposed attention modules. The model is optimized using AdamW [39] with an initial learning rate of $1 \times 10^{-5}$, which is reduced by a factor of $0.1$ at epoch 10 and 15. The training process converges within 20 epochs, with each epoch comprising 500 sampled episodes. Input images are resized to $448 \times 448$ resolution without data augmentation. We set the number of learnable proxies ($\mathcal{P}$) to $M = 25$ by default and use a loss balance weight ($\lambda$) of 1.0. Following previous works [23, 31], we set the number of normal references as $K_1 \in [1, 2, 4]$. Considering the scarcity of abnormal samples, we only use one abnormal reference ($K_2 = 1$), making our approach highly applicable in real-world scenarios. The implementation is based on PyTorch@2.1.1, and the experiments are conducted on a single NVIDIA RTX 4090 24GB GPU. To ensure statistical reliability, we report results averaged across 3 independent runs with different random seeds.

Table 2: The image/pixel-level AUROC scores of the proposed method and previous methods on the BraTS dataset. Results marked with ∗ are based on our re-implementation. Other results are reported from [60].

| Setting | Method | Image | Pixel | Mean |
|---|---|---|---|---|
| $N^1$ | ResAD∗ [60] | 73.5 | 91.0 | 82.3 |
| $N^1 + A^1$ | **Ours** | **78.1** | **96.5** | **87.3** |
| | SPADE [10] | 58.0 | 92.8 | 75.4 |
| | PaDiM [13] | 59.4 | 90.2 | 74.8 |
| | PatchCore [48] | 58.2 | 93.5 | 75.9 |
| $N^2$ | RegAD [21] | 54.6 | 81.4 | 68.0 |
| | WinCLIP [23] | 55.9 | 91.5 | 73.7 |
| | InCTRL [68] | 74.6 | - | - |
| | ResAD∗ [60] | 66.2 | 91.5 | 78.9 |
| $N^2 + A^1$ | **Ours** | **82.1** | **96.8** | **89.5** |
| | SPADE [10] | 66.3 | 94.8 | 80.6 |
| | PaDiM [13] | 60.6 | 94.5 | 77.6 |
| | PatchCore [48] | 71.2 | 95.9 | 83.6 |
| $N^4$ | RegAD [21] | 60.0 | 87.3 | 73.7 |
| | WinCLIP [23] | 67.3 | 93.2 | 80.3 |
| | MVFA∗ [22] | 75.2 | 92.7 | 84.0 |
| | InCTRL [68] | 76.9 | - | - |
| | ResAD∗ [60] | 74.9 | 94.3 | 84.6 |
| $N^4 + A^1$ | **Ours** | **84.9** | **97.1** | **91.0** |

Table 3: Ablation study on the MVTecAD and VisA datasets. Image/pixel-level AUROC are reported. $N^1/A^1$ represents the one normal/abnormal reference sample, "NAGL" indicates the use of our proposed framework. There are four different settings. i/ii uses only normal/abnormal samples through NN search, iii merges the results of i and ii without additional processing, and iv includes our proposed process.

| | $N^1$ | $A^1$ | NAGL | MVTecAD | | VisA | | Mean |
|---|---|---|---|---|---|---|---|---|
| | | | | Image | Pixel | Image | Pixel | |
| i | ✓ | | | 93.2 | 94.5 | 81.5 | 95.3 | 91.1 |
| ii | | ✓ | | 70.1 | 83.3 | 58.8 | 84.5 | 74.2 |
| iii | ✓ | ✓ | | 90.7 | 92.1 | 77.2 | 93.5 | 88.4 |
| iv | ✓ | ✓ | ✓ | **95.8** | **96.6** | **88.5** | **97.5** | **94.6** |

Table 4: Comparison of the total parameters, training time, and inference speed of the proposed method with InCTRL and ResAD. The training time of InCTRL and ResAD is measured under the $N^1$ setting, while our method is based on the $N^1 + A^1$ setting.

| Method | Total Parameters (M) | Training Time (H) | Inference Speed (FPS) |
|---|---|---|---|
| InCTRL | 117.5 | 0.7 | 1.2 |
| ResAD | 59.2 | 20.6 | 7.8 |
| Ours | **24.4** | **0.3** | **17.1** |

**Comparison Methods.** We select some representative few-normal-shot AD methods to comparison, including SPADE [10], PaDiM [13], PatchCore [48], RegAD [21], WinCLIP [23], MVFA [22], and PromptAD [31]. For generalist AD, we primarily compared our approach with InCTRL [68] and ResAD [60]. Tab. 1 and Tab. 2 present the comparative results of our method against these baselines. The results marked with † are reported by [23]. Due to incomplete ResAD results (only reporting 2/4-shot settings with AUROC metrics), we reproduce the results using their official code (marked with ∗). For further details, please refer to Appendix D.

## 4.2 Main Results

Tab. 1 and Tab. 2 present comparative results between our proposed method and existing approaches across MVTecAD, VisA, and BraTS datasets. The experimental results demonstrate three key advantages of our method. Additionally, to demonstrate the generalization capabilities, we also present the performance on MVTec3D [4], MVTecLOCO [1], BTAD [46], and MPDD [24] datasets in Appendix G.

**Benefits of Single Abnormal Sample.** Incorporating just one abnormal reference sample ($A^1$) yields substantial performance gains across all scenarios. On industrial datasets (Tab. 1), our $N^1 + A^1$ setting achieves 95.8% and 96.6% for image and pixel AUROC, respectively on MVTecAD, surpassing the best $N^4$ baseline method (WinCLIP: 95.2%, 96.2%) by **0.6** and **0.4** percentage points while using fewer reference samples. A similar trend is observed on VisA, where our $N^2 + A^1$ setting achieves 89.8% and 97.6% for image and pixel AUROC, outperforming the best baseline method (ResAD with $N^4$: 89.3%, 96.8%) by **0.5** and **0.8** percentage points, respectively. These results show significant benefits from abnormal reference data.

**Generalization.** Our method exhibits superior generalization capabilities compared to ResAD across multiple datasets. On MVTecAD, under the $N^1 + A^1$ setting, our approach achieves 95.7% image F1-max and 58.9% pixel F1-max, outperforming ResAD by **4.5** and **10.7** percentage points, respectively. Similarly, on VisA, we attain 85.4% image F1-max and 41.8% pixel F1-max, exceeding ResAD by **4.1** and **4.0** percentage points. Furthermore, as shown in Tab. 2, our method demonstrates superior cross-domain generalization, achieving 89.5% average AUROC on the BraTS medical dataset with just two normal and one abnormal sample, a **10.6** percentage point improvement over ResAD

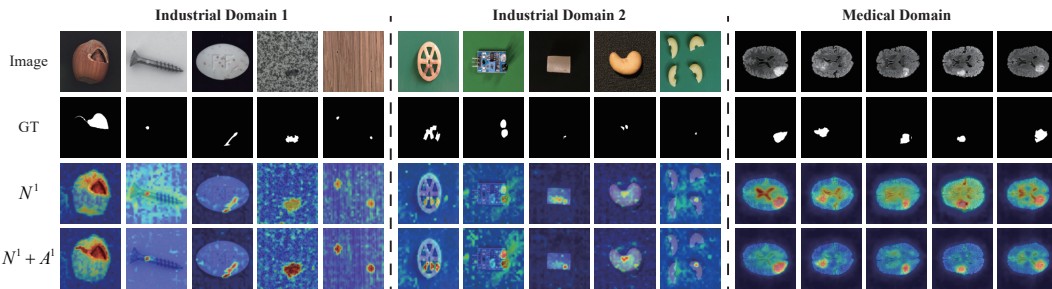

Figure 3: Qualitative results. The first row displays the input images, and the second row shows the ground truth. The third row illustrates anomaly score maps using 1 normal sample, while the bottom row shows anomaly score maps produced by our method using 1 normal and 1 abnormal samples.

(78.9% with $N^2$ setting). Notably, across all other experimental settings, our method consistently outperforms existing approaches in both anomaly detection and segmentation. These results clearly demonstrate the effectiveness of our method in generalist AD.

**Scalability.** Through comparative analysis, our method demonstrates excellent scalability as the number of normal samples increases across all three datasets. For example, on VisA, we observe consistent performance improvements when progressing from $N^1 + A^1$ to $N^2 + A^1$ and $N^4 + A^1$ settings. Specifically, the image F1-max score increases from 85.4% to 86.8% and 87.8%, while the pixel F1-max score improves from 41.8% to 43.3% and reaches 44.0%. On BraTS, the average AUROC improves from 87.3% to 89.5% and reaches 91.0%. These results indicate that our method effectively utilizes additional normal samples to enhance the detection ability.

## 4.3 Ablation Study

To further investigate the effectiveness of our method, we conduct an ablation study on the MVTecAD and VisA datasets. Tab. 3 presents the results from different experimental settings. The results clearly demonstrate several important findings.

First, comparing the results of i, ii, and iii, we observe that naively incorporating abnormal references leads to significant performance degradation both at the image and pixel levels. This aligns with our argument in Sec. 1 that the abnormal features cause severe false activation, where the unrefined abnormal features introduce misleading guidance that incorrectly highlights normal patterns as anomalies, thereby compromising detection accuracy.

Second, when comparing settings iii and iv, the benefits of our proposed method become evident. By implementing the NAGL framework in setting iv, our method manages to effectively mitigate the false activation problem observed when naively using an abnormal reference. Properly mining the abnormal variations in residual space to guide detection can effectively suppress the over-activation, resulting in 6.2 average AUROC improvements from 88.4% to 94.6%.

Finally, the comparison between i and iv reveals that our method fully unleashes the potential of abnormal references, with gains of +2.6 Image AUROC on MVTecAD and +7.0 on VisA, proving that properly processed abnormal samples provide complementary discriminative signals.

## 4.4 Analysis of two attentions in RM-AFL module

In the first attention (RM), we set the value as $\text{Res}(\mathcal{F}^a, \mathcal{F}^n)$, which provides the *normal-abnormal differences* in the residual space. We define the key as $\mathcal{F}^a$, which are the original abnormal patterns. Since the query is learnable, the output is the optimally aggregated residual features (termed as residual proxies $\tilde{\mathcal{P}}$).

The second attention (AFL) aims to learn query-related abnormal patterns by comparing *normal-abnormal differences* and *normal-query differences*. We set the query as $\tilde{\mathcal{P}}$ and the key as $\text{Res}(\mathcal{F}^q, \mathcal{F}^n)$. This *comparison* is achieved by computing the attention map between $\tilde{\mathcal{P}}$ and

$\text{Res}(\mathcal{F}^q, \mathcal{F}^n)$. Then the query features $\mathcal{F}^q$ are aggregated by the attention map to obtain the query-related abnormal patterns (termed as anomaly proxies $\hat{\mathcal{P}}$).

The residual proxies $\tilde{\mathcal{P}}$ represent abnormal patterns of reference abnormal samples in residual space, and anomaly proxies $\hat{\mathcal{P}}$ represent abnormal patterns of the query sample in vision space. As validated in Appendix B, residual features exhibit a common distribution, even among residual features of different anomalies. Residual features of known anomalies can provide references; if the residual feature of a query is similar to them, the corresponding visual feature is likely to be an anomaly.

### 4.5 Efficiency Analysis

We compare the efficiency of our method with InCTRL and ResAD in terms of total parameters, training time, and inference speed. Tab. 4 shows that our method is more efficient than InCTRL and ResAD regarding total parameters and training time. Our method has 24.4 million parameters, which is around **5× smaller** than InCTRL. Our method only requires 0.3 hours for training, which is **69× faster** than ResAD. In terms of inference speed, our method achieves 17.1 FPS, which is **14× faster** than InCTRL and **2× faster** than ResAD. These results demonstrate the efficiency of our method for real-world deployment.

### 4.6 Qualitative Results

Fig. 3 presents qualitative results from both industrial and medical datasets. Comparing the score maps, we observe that our method effectively generates more accurate results, which demonstrate that incorporating abnormal reference samples provides valuable guidance boundaries for the detection process, and our method effectively leverages this information to improve detection accuracy.

## 5 Conclusion

In this paper, we introduce a novel task for anomaly detection using both normal and abnormal references. Our approach addresses the limitations of traditional methods by leveraging limited known anomalies to guide the detection of unseen anomalies. We propose a NAGL framework that extracts discriminative features in the residual space, effectively capturing the essence of anomalies. Experimental results across multiple benchmarks demonstrate that our method significantly outperforms existing approaches, particularly in detecting challenging unseen anomalies. The performance gains are consistent across various datasets and settings, highlighting the robustness of our approach. This work opens new research directions for anomaly detection with limited supervision, with potential applications in industrial inspection and medical diagnosis. Future research could delve into more effective methods for leveraging scarce anomaly samples or expanding prompts. For instance, it could involve enabling language-based descriptions for normal or abnormal references.

## 6 Acknowledgements

This work is supported by the National Natural Science Foundation of China (No. 62471405, 62331003, 62301451), Jiangsu Basic Research Program Natural Science Foundation (BK20241814), Suzhou Basic Research Program (SYG202316) and XJTLU REF-22-01-010, XJTLU AI University Research Center, Jiangsu Province Engineering Research Center of Data Science and Cognitive Computation at XJTLU and SIP AI innovation platform (YZCXPT2022103).

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

# A  Dataset Collection

To implement our normal-abnormal-guided generalist anomaly detection, we construct new benchmarks using three popular public datasets, including MVTecAD [2], VisA [69], and BraTS [45]. We conduct evaluations from two perspectives: (1) industrial-to-industrial evaluation across different industrial scenarios (MVTecAD ↔ VisA), and (2) industrial-to-medical evaluation across industrial and medical domains (MVTecAD → BraTS). Specifically, when MVTecAD serves as the training set, we test on VisA and BraTS; when testing on MVTecAD, we use VisA as the training set.

**Original Dataset Structure.** Each dataset is individually divided into its own training and test sets. For all datasets, their respective training sets contain only *good* samples, while each dataset has a unique structure for its corresponding test set. As shown in Tab. 5. MVTecAD contains 15 object classes with 73 different defect types, distinguishing between various defect categories. VisA includes 12 object classes with various defect types, but all anomalies within each object class are grouped together without specific categorization. BraTS exclusively comprises brain MRI scans, with abnormal samples including *tumour* segmentation.

Table 5: Statistics of the datasets. $|\mathcal{C}|$ denotes the number of classes, $|\mathcal{T}|$ denotes the number of defect types, $N$ and $A$ represent the normal and abnormal samples, respectively. $K_1/K_2$ denotes the number of normal/abnormal reference samples.

| Dataset | $|\mathcal{C}|$ | $|\mathcal{T}|$ | Test $N$ | Test $A$ | Reference $N$ | Reference $A$ |
|---------|---------|---------|------|------|--------------|--------------|
| MVTecAD | 15 | 73 | 467 | 1258 | $15 \times K_1$ | $73 \times K_2$ |
| VisA | 12 | - | 962 | 1200 | $12 \times K_1$ | $12 \times K_2$ |
| BraTS | 1 | 1 | 154 | 1097 | $1 \times K_1$ | $1 \times K_2$ |

**Our Input Data.** During both training and testing, we implement a random sampling strategy to construct our input data. Specifically, our input data consists of a query input and a reference set that includes both normal and abnormal samples. We first randomly select a sample from the entire test set as the query input. Then, based on the query input's category, we randomly select $K_1$ normal samples from the training set as normal references. Additionally, according to the query input's *defect-type*, we randomly select $K_2$ abnormal samples as abnormal references. During training, each episode input are randomly combined samples. Unlike the training process, during testing, we randomly selected reference set only once for each anomaly type and then use this set to test all samples of this anomaly type. The detailed statistics of our reference set are shown in Tab. 5.

**Defect type between Query and Reference.** As shown in Fig. 4, there are some input cases: (a) the input query and abnormal reference samples belong to the same defect type; (b) the input query and abnormal reference samples exhibit different defect types; (c) when the input query is normal, the abnormal reference randomly selects one defect type. The MVTecAD dataset categorizes different defect types separately, thereby ensuring that the input query ($\mathbf{x}^q$) and abnormal reference samples ($\mathcal{R}^a$) belong to the same defect category (Fig. 4a). In contrast, the VisA dataset combines all anomaly samples for each product type without distinguishing defect types, which results in cases where the input query

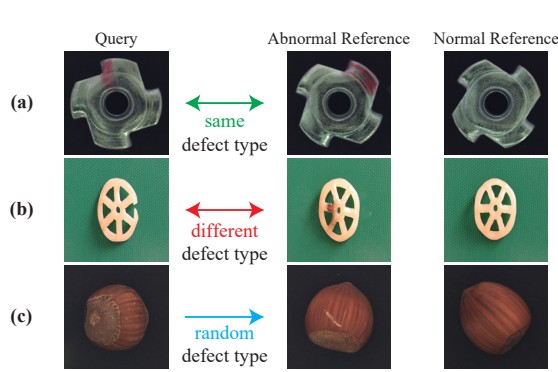

Figure 4: Some examples of our input data

and abnormal reference may exhibit different types of anomalies (Fig. 4b). Additionally, when the input query is normal, the abnormal reference randomly selects one defect type (Fig. 4c).

# B  Residual Features

To explore the guidance of abnormal residual features in anomaly detection, we visualize the feature distribution in the original vision space and the residual feature space. As shown in Fig. 5, the features from different defect types are significantly different in the original vision space, while these features are more concentrated in their distribution in the residual feature space, suggesting it provides a unified cross-domain representation for anomalies. Moreover, as shown in Fig. 6, the

L2-norm of normal and abnormal features are overlapped in the original vision space, while they are separated in the residual feature space, indicating that residual features offer a more discriminative attribute for distinguishing between normal and abnormal samples.

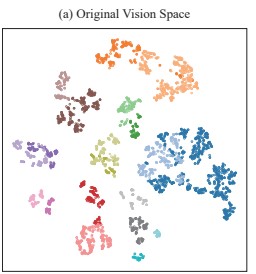
(a) Original Vision Space

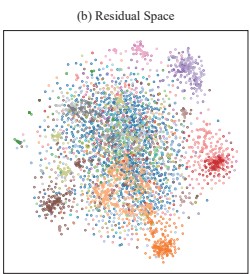
(b) Residual Space

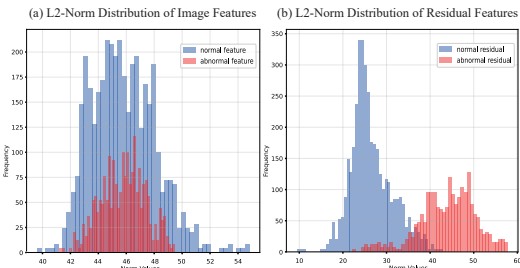
(a) L2-Norm Distribution of Image Features     (b) L2-Norm Distribution of Residual Features

Figure 5: T-SNE visualization of features. Different colours denote different defect types. (a) In the original vision space, the features from different defect types are significantly different. (b) In the residual feature space, these features are more overlapped in their distribution.

Figure 6: (a) In the original visual feature space, the L2-norm distributions of normal and anomalous features exhibit significant overlap. (b) In the residual feature space, the L2-norm distributions of normal and anomalous features are more clearly separated.

## C    More Abnormal References

Considering the scarcity of anomalous samples in real-world scenarios, we focused on using only one anomalous reference sample in the main text. To further explore the scalability of the NAGL framework, we conducted experiments using two anomalous reference samples. As shown in Table 6, using two anomalous reference samples improved the performance of the NAGL framework on both MVTecAD and VisA datasets. This shows NAGL effectively uses multiple anomalous references to improve detection performance.

Table 6: The AUROC performance comparison between 1 and 2 abnormal references ($A^1$ and $A^2$).

| Setting | MVTecAD | VisA | Mean | $\Delta$ |
|---|---|---|---|---|
| $N^1 + A^1$ | 95.8 | 88.5 | 92.2 | |
| $N^1 + A^2$ | 96.5 | 89.5 | 93.0 | + 0.8 |
| $N^2 + A^1$ | 96.8 | 89.8 | 93.3 | |
| $N^2 + A^2$ | 97.3 | 90.5 | 93.9 | + 0.6 |
| $N^4 + A^1$ | 97.1 | 91.2 | 94.2 | |
| $N^4 + A^2$ | 97.5 | 92.2 | 94.9 | + 0.7 |

## D    Compared with ViT-based Results

Due to the extensive training time required for ResAD, we only reproduced the CNN-based (WideResNet50) results in the main text, maintaining identical training hyperparameters as specified in the original paper. As shown in Tab. 7, for a more equitable comparison, we also compare our approach with ViT-based results. It is clear that our method still achieves competitive performance.

Table 7: The AUROC performance comparison of ViT-based backbones on MVTecAD and VisA datasets.

| Setting | Method | MVTecAD Image | MVTecAD Pixel | VisA Image | VisA Pixel | Setting | Method | MVTecAD Image | MVTecAD Pixel | VisA Image | VisA Pixel |
|---|---|---|---|---|---|---|---|---|---|---|---|
| $N^2$ | WinCLIP | 94.4 | 96.0 | 84.6 | 96.8 | $N^4$ | WinCLIP | 95.2 | 96.2 | 87.3 | 97.2 |
| | InCTRL | 94.0 | - | 85.8 | - | | InCTRL | 94.5 | - | 87.7 | - |
| | ResAD | 94.4 | 95.6 | 84.5 | 95.1 | | ResAD | 94.2 | 96.9 | 90.8 | 97.5 |
| $N^2 + A^1$ | Ours | 96.8 | 96.8 | 89.8 | 97.6 | $N^4 + A^1$ | Ours | 97.1 | 97.0 | 91.2 | 97.8 |

## E    Discussion

**Limitations:** A drawback of our study is that it focuses solely on image data for experimentation. It would be highly beneficial to apply our approach to other data types, like video and time series, to thoroughly assess the adaptability of our method.

**Social Impacts:** As a unified framework for generalist anomaly detection, the introduced approach does not raise specific ethical issues or adverse societal effects. The datasets utilized are publicly available. All qualitative illustrations are derived from industrial product imagery, ensuring no violation of personal privacy.

## F  Hyperparameter Analysis

Fig. 7 shows the impact of the number of learnable proxies $M$. We observe that the performance of our method increases as $M$ increases. This is because a larger number of proxies can better capture the diversity of the feature space, which is beneficial for the refinement process. However, the performance improvement diminishes as $M$ exceeds 25. One possible reason is the model becomes over-parameterized, leading to overfitting. Therefore, we set $M = 25$ in our experiments to balance performance and efficiency.

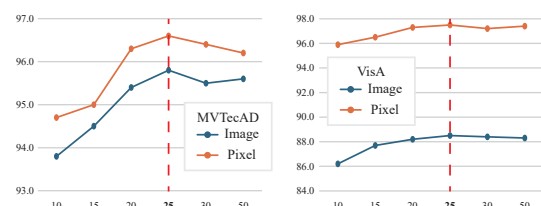

Figure 7: Impact of the number of learnable proxies $M$ on MVTecAD and VisA datasets.

## G  Additional Results

Additionally, we evaluated the performance of our proposed method on the MVTec3D and MVTecLOCO datasets to further investigate its generalizability. As shown in Tab. 8, our method achieves competitive results compared to ResAD in both 3D anomaly detection and logical anomaly detection, demonstrating its generalizability.

Table 8: Comparison of the proposed method with the ResAD on MVTec3D and MVTecLOCO datasets.

| Setting | Method | MVTec3D | | | | | | MVTecLOCO | | | | | |
|---|---|---|---|---|---|---|---|---|---|---|---|---|---|
| | | Image-level | | | Pixel-level | | | Image-level | | | Pixel-level | | |
| | | AUROC | AP | F1-max | AUROC | PRO | F1-max | AUROC | AP | F1-max | AUROC | PRO | F1-max |
| $N^1$ | ResAD* | 63.8 | 86.4 | 88.7 | 94.2 | 88.2 | 21.2 | 60.6 | 73.9 | 77.8 | 65.9 | 60.8 | 17.9 |
| $N^1 + A^1$ | Ours | **83.0** | **95.0** | **91.7** | **94.7** | **98.4** | **50.3** | **63.3** | **76.3** | **78.4** | **66.6** | **69.9** | **19.4** |
| $N^2$ | ResAD* | 66.7 | 88.6 | 88.9 | **94.9** | 90.1 | 24.5 | 62.3 | 76.3 | 77.7 | 66.0 | 62.3 | 18.3 |
| $N^2 + A^1$ | Ours | **82.8** | **94.9** | **91.6** | 94.8 | **98.4** | **51.3** | **64.6** | **77.4** | **78.0** | **66.3** | **69.3** | **18.5** |
| $N^4$ | ResAD* | 70.1 | 89.4 | 88.9 | 95.0 | 91.4 | 26.0 | 65.7 | 77.5 | 77.7 | 67.5 | 60.7 | **19.1** |
| $N^4 + A^1$ | Ours | **86.9** | **96.4** | **92.3** | **95.4** | **98.6** | **54.2** | **71.5** | **81.5** | **79.1** | 66.9 | **69.2** | 18.6 |

As shown in Fig. 8, we provide further qualitative results obtained from our NAGL for pixel-level anomaly detection. The results demonstrate that our approach accurately localizes both large and small surface defects across various test cases. Furthermore, we report the detailed subset-level results ($mean \pm std$) of NAGL on MVTecAD, VisA, BraTS, MVTec3D and MVTecLOCO datasets, under $N^1 + A^1$ (Tab. 9), $N^2 + A^1$ (Tab. 10), and $N^4 + A^1$ (Tab. 11) settings. We also report performance on the BTAD and MPDD datasets.

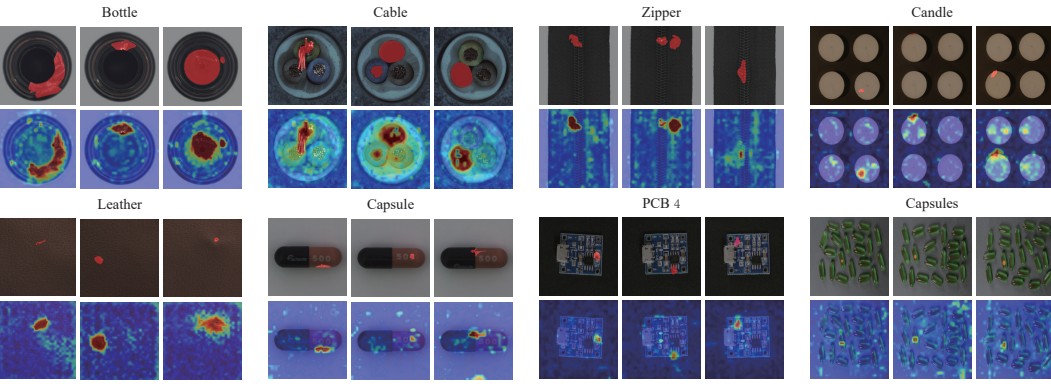

Figure 8: Qualitative results for eight subcategories with different defect types.

Table 9: The comprehensive results of our NAGL framework, evaluated under settings with $N^1 + A^1$.

| Dataset | Objects | Image-level | | | Pixel-level | | |
|---|---|---|---|---|---|---|---|
| | | AUROC | AP | F1-max | AUROC | PRO | F1-max |
| **MVTecAD** | bottle | 99.2±0.9 | 99.7±0.4 | 99.2±0.0 | 98.2±0.1 | 95.8±0.2 | 76.2±0.8 |
| | cable | 93.1±0.9 | 96.3±0.6 | 90.4±2.0 | 94.2±0.6 | 87.8±0.5 | 58.9±0.8 |
| | capsule | 91.5±7.1 | 98.1±1.7 | 95.5±0.7 | 98.1±0.2 | 96.7±0.4 | 46.8±5.3 |
| | carpet | 100.0±0.0 | 100.0±0.0 | 100.0±0.0 | 99.3±0.0 | 98.5±0.0 | 68.9±0.6 |
| | grid | 99.8±0.2 | 99.9±0.1 | 99.4±0.5 | 99.4±0.0 | 96.9±0.3 | 51.4±0.9 |
| | hazelnut | 99.5±0.4 | 99.7±0.2 | 98.3±1.1 | 99.5±0.1 | 97.1±0.4 | 77.5±2.7 |
| | leather | 100.0±0.0 | 100.0±0.0 | 100.0±0.0 | 99.1±0.1 | 98.5±0.3 | 41.3±1.4 |
| | metal_nut | 99.4±0.1 | 99.9±0.0 | 98.2±0.3 | 95.2±0.3 | 93.0±0.5 | 68.5±1.2 |
| | pill | 93.6±1.6 | 98.7±0.3 | 95.9±0.5 | 94.2±0.3 | 96.2±0.1 | 52.9±2.3 |
| | screw | 77.2±0.9 | 91.5±0.4 | 86.7±0.2 | 97.8±0.4 | 92.2±0.8 | 43.2±2.4 |
| | tile | 99.8±0.1 | 99.9±0.0 | 99.0±0.3 | 96.1±0.4 | 91.4±0.3 | 70.1±0.4 |
| | toothbrush | 99.4±0.2 | 99.8±0.0 | 98.3±0.0 | 99.2±0.1 | 95.1±0.2 | 65.9±2.2 |
| | transistor | 86.1±4.2 | 79.3±6.9 | 76.7±4.1 | 84.3±3.7 | 64.7±1.7 | 39.1±6.4 |
| | wood | 99.8±0.1 | 99.9±0.0 | 98.9±0.5 | 96.0±0.9 | 95.8±0.5 | 67.9±0.7 |
| | zipper | 99.1±0.2 | 99.8±0.1 | 98.5±0.5 | 97.9±0.1 | 94.0±0.3 | 54.5±0.6 |
| | **Mean** | **95.8±0.7** | **97.5±0.6** | **95.7±0.3** | **96.6±0.2** | **92.9±0.0** | **58.9±0.5** |
| **VisA** | candle | 93.9±0.7 | 94.3±0.3 | 87.3±1.0 | 99.0±0.0 | 97.1±0.3 | 37.6±0.9 |
| | capsules | 96.7±0.4 | 98.0±0.2 | 93.5±0.1 | 97.9±0.4 | 96.4±0.4 | 44.4±4.4 |
| | cashew | 88.5±2.3 | 94.3±0.9 | 87.2±1.8 | 98.7±0.2 | 97.6±1.1 | 56.7±1.5 |
| | chewinggum | 97.3±0.3 | 98.9±0.1 | 95.7±0.3 | 99.3±0.1 | 84.6±0.6 | 68.7±0.4 |
| | fryum | 95.7±0.7 | 98.1±0.3 | 92.8±0.9 | 95.8±0.4 | 91.1±0.4 | 40.8±2.4 |
| | macaroni1 | 89.2±2.6 | 88.7±3.1 | 83.1±2.3 | 99.4±0.0 | 94.1±1.0 | 27.6±2.9 |
| | macaroni2 | 59.5±7.5 | 55.9±6.3 | 68.3±1.0 | 98.2±0.6 | 86.2±2.5 | 12.6±4.2 |
| | pcb1 | 94.9±0.7 | 93.4±0.7 | 90.6±1.2 | 98.9±0.2 | 94.2±0.1 | 60.7±4.6 |
| | pcb2 | 86.3±0.9 | 84.3±1.4 | 80.0±1.4 | 95.6±0.1 | 86.1±0.3 | 35.5±1.0 |
| | pcb3 | 85.6±2.1 | 88.5±1.6 | 78.5±0.7 | 93.7±0.1 | 86.6±0.1 | 42.0±2.5 |
| | pcb4 | 79.1±11.1 | 80.3±7.4 | 76.3±9.7 | 94.9±0.6 | 81.9±3.7 | 25.7±2.2 |
| | pipe_fryum | 95.8±0.6 | 98.0±0.3 | 91.9±1.0 | 98.8±0.1 | 97.8±0.0 | 48.9±2.2 |
| | **Mean** | **88.5±1.4** | **89.4±1.2** | **85.4±0.7** | **97.5±0.1** | **91.1±0.4** | **41.8±0.5** |
| **BraTS** | **brain** | **78.1±3.4** | **95.4±0.9** | **93.8±0.3** | **96.5±0.1** | **79.4±0.5** | **49.4±1.0** |
| **MVTec3D** | bagel | 95.5±2.0 | 98.9±0.5 | 95.9±1.6 | 98.9±0.1 | 99.7±0.1 | 66.9±0.8 |
| | cable_gland | 77.2±11.1 | 93.4±4.2 | 90.4±0.7 | 95.6±2.2 | 98.4±0.8 | 38.5±20.1 |
| | carrot | 87.4±5.0 | 96.8±1.5 | 93.0±1.3 | 99.1±0.2 | 99.8±0.0 | 54.2±2.8 |
| | cookie | 83.9±6.1 | 95.3±1.8 | 90.5±1.9 | 92.9±0.3 | 97.7±0.2 | 57.7±4.2 |
| | dowel | 74.8±2.7 | 92.9±1.2 | 90.7±1.0 | 94.3±0.5 | 98.6±0.2 | 40.0±3.5 |
| | foam | 78.0±2.2 | 94.6±0.5 | 88.9±0.0 | 79.8±1.2 | 92.8±0.7 | 43.5±1.7 |
| | peach | 93.8±1.8 | 98.0±0.9 | 94.9±0.8 | 99.2±0.1 | 99.8±0.0 | 62.5±4.4 |
| | potato | 67.7±8.5 | 89.7±3.6 | 89.9±1.0 | 98.1±0.3 | 99.5±0.1 | 42.5±4.6 |
| | rope | 96.2±1.3 | 98.6±0.4 | 94.7±0.8 | 97.7±0.0 | 99.4±0.0 | 50.4±0.2 |
| | tire | 75.3±5.8 | 92.5±1.8 | 87.9±0.4 | 91.9±0.3 | 98.2±0.1 | 46.4±0.9 |
| | **Mean** | **83.0±1.5** | **95.0±0.6** | **91.7±0.4** | **94.7±0.2** | **98.4±0.1** | **50.3±2.4** |
| **MVTecLOCO** | breakfast_box | 77.1±2.8 | 87.1±1.7 | 79.3±1.8 | 67.1±1.6 | 79.7±1.2 | 32.3±0.9 |
| | juice_bottle | 50.8±2.1 | 74.6±1.9 | 83.5±0.3 | 71.4±0.5 | 85.9±0.4 | 30.8±1.7 |
| | pushpins | 62.7±0.6 | 66.4±1.4 | 73.5±1.9 | 53.1±1.4 | 54.3±1.4 | 3.1±0.1 |
| | screw_bag | 59.7±1.1 | 75.8±1.0 | 78.5±0.3 | 67.8±1.6 | 63.7±1.2 | 10.8±0.1 |
| | splicing_connectors | 66.2±5.3 | 77.5±6.1 | 76.9±0.4 | 73.7±1.3 | 65.7±3.5 | 20.0±1.9 |
| | **Mean** | **63.3±1.2** | **76.3±1.0** | **78.4±0.5** | **66.6±0.3** | **69.9±1.0** | **19.4±0.2** |
| **BTAD** | 01 | 85.9±3.0 | 95.0±1.1 | 86.9±2.0 | 65.4±1.7 | 93.3±1.0 | 50.9±2.2 |
| | 02 | 93.6±1.4 | 99.0±0.2 | 95.6±0.6 | 68.4±5.8 | 96.9±0.4 | 64.6±3.2 |
| | 03 | 99.8±0.0 | 96.7±0.5 | 93.2±2.1 | 97.3±0.3 | 99.5±0.0 | 65.6±1.6 |
| | **Mean** | **93.1±1.3** | **96.9±0.5** | **91.9±1.3** | **77.1±2.0** | **96.6±0.3** | **60.4±0.7** |
| **MPDD** | bracket_black | 55.7±2.9 | 66.6±6.5 | 76.0±1.6 | 92.0±1.2 | 95.3±0.6 | 23.1±5.8 |
| | bracket_brown | 57.6±4.2 | 75.0±3.8 | 81.6±0.7 | 87.0±1.3 | 94.9±0.2 | 10.9±0.3 |
| | bracket_white | 76.2±2.8 | 68.9±7.3 | 84.6±2.1 | 95.9±0.7 | 99.7±0.0 | 24.7±4.1 |
| | connector | 87.1±3.2 | 70.9±6.1 | 76.4±3.9 | 91.2±1.5 | 97.4±0.5 | 28.8±4.7 |
| | metal_plate | 99.9±0.0 | 100.0±0.0 | 99.3±0.0 | 89.7±0.9 | 96.5±0.5 | 73.2±1.7 |
| | tubes | 94.5±1.1 | 97.8±0.6 | 92.0±0.5 | 95.5±0.5 | 98.8±0.1 | 64.6±1.5 |
| | **Mean** | **78.5±0.8** | **79.9±1.1** | **85.0±0.5** | **91.9±0.6** | **97.1±0.2** | **37.5±2.5** |

Table 10: The comprehensive results of our NAGL framework, evaluated under settings with $N^2+A^1$.

| Dataset | Objects | Image-level | | | Pixel-level | | |
|---|---|---|---|---|---|---|---|
| | | AUROC | AP | F1-max | AUROC | PRO | F1-max |
| **MVTecAD** | bottle | 99.2±1.0 | 99.7±0.4 | 99.5±0.5 | 98.2±0.2 | 96.0±0.5 | 76.5±1.4 |
| | cable | 93.7±0.4 | 96.6±0.3 | 90.8±0.5 | 94.7±0.1 | 88.5±0.1 | 60.0±1.4 |
| | capsule | 90.8±8.3 | 97.8±2.1 | 95.4±1.3 | 98.1±0.1 | 96.8±0.5 | 47.4±5.2 |
| | carpet | 100.0±0.0 | 100.0±0.0 | 100.0±0.0 | 99.3±0.0 | 98.5±0.0 | 68.9±0.4 |
| | grid | 99.9±0.1 | 100.0±0.0 | 99.7±0.5 | 99.4±0.0 | 97.0±0.0 | 51.3±0.8 |
| | hazelnut | 99.8±0.1 | 99.9±0.1 | 99.3±0.0 | 99.6±0.1 | 97.4±0.4 | 79.8±1.0 |
| | leather | 100.0±0.0 | 100.0±0.0 | 100.0±0.0 | 99.1±0.0 | 98.4±0.3 | 41.8±1.0 |
| | metal_nut | 99.8±0.1 | 99.9±0.0 | 99.3±0.3 | 95.5±0.3 | 93.5±0.5 | 70.2±1.1 |
| | pill | 95.5±1.6 | 99.1±0.4 | 96.6±0.2 | 94.3±0.3 | 96.2±0.2 | 54.4±1.1 |
| | screw | 86.0±0.9 | 95.2±0.4 | 88.6±1.6 | 98.2±0.2 | 93.5±0.3 | 48.8±0.4 |
| | tile | 99.9±0.1 | 100.0±0.0 | 99.6±0.3 | 96.1±0.3 | 91.4±0.4 | 69.9±0.4 |
| | toothbrush | 99.4±0.7 | 99.7±0.3 | 98.4±1.6 | 99.2±0.1 | 95.5±0.2 | 65.5±2.0 |
| | transistor | 88.0±4.1 | 80.8±4.6 | 79.2±4.7 | 85.2±1.2 | 66.0±1.8 | 39.0±4.0 |
| | wood | 99.8±0.1 | 99.9±0.0 | 99.2±0.0 | 96.4±0.8 | 95.9±0.3 | 68.8±1.4 |
| | zipper | 99.5±0.2 | 99.9±0.0 | 98.9±0.2 | 98.0±0.2 | 94.1±0.4 | 54.7±0.7 |
| | **Mean** | **96.8±0.8** | **97.9±0.4** | **96.3±0.5** | **96.8±0.2** | **93.2±0.2** | **59.8±0.3** |
| **VisA** | candle | 93.7±0.7 | 94.2±0.5 | 87.9±0.5 | 99.0±0.0 | 97.1±0.2 | 37.9±1.2 |
| | capsules | 96.9±1.0 | 98.2±0.5 | 93.4±1.7 | 97.9±0.2 | 96.5±0.4 | 48.0±3.7 |
| | cashew | 90.1±3.6 | 95.3±1.7 | 88.9±2.6 | 98.7±0.1 | 97.8±0.8 | 57.2±0.5 |
| | chewinggum | 97.7±0.5 | 99.0±0.2 | 95.5±0.4 | 99.3±0.1 | 85.1±1.3 | 68.1±0.7 |
| | fryum | 95.9±0.1 | 98.2±0.1 | 94.0±0.2 | 95.7±0.2 | 91.1±0.3 | 41.3±1.2 |
| | macaroni1 | 89.1±0.3 | 88.5±0.8 | 83.4±0.5 | 99.4±0.0 | 93.8±0.6 | 27.1±1.8 |
| | macaroni2 | 63.2±5.4 | 59.3±4.3 | 68.8±1.4 | 98.3±0.2 | 86.7±0.4 | 17.3±3.8 |
| | pcb1 | 94.8±0.8 | 92.8±1.2 | 91.8±0.4 | 99.0±0.1 | 94.3±0.1 | 61.1±1.8 |
| | pcb2 | 86.6±1.6 | 84.4±1.4 | 81.5±2.4 | 95.7±0.1 | 86.5±0.1 | 36.9±0.8 |
| | pcb3 | 88.9±1.1 | 91.2±0.7 | 83.1±1.8 | 93.9±0.1 | 87.3±0.0 | 44.1±2.5 |
| | pcb4 | 84.6±10.7 | 87.4±8.3 | 80.0±9.6 | 95.2±0.5 | 83.2±4.0 | 29.7±1.8 |
| | pipe_fryum | 96.5±1.2 | 98.3±0.6 | 93.1±0.8 | 98.9±0.1 | 97.8±0.1 | 50.7±1.5 |
| | **Mean** | **89.8±1.5** | **90.6±0.9** | **86.8±1.1** | **97.6±0.0** | **91.4±0.4** | **43.3±0.3** |
| **BraTS** | **brain** | **82.1±3.0** | **96.6±1.0** | **93.9±0.1** | **96.8±0.2** | **79.9±0.4** | **51.8±1.5** |
| **MVTec3D** | bagel | 95.6±2.6 | 98.9±0.7 | 95.3±1.8 | 99.0±0.1 | 99.6±0.0 | 67.8±1.0 |
| | cable_gland | 82.0±4.7 | 95.4±1.1 | 91.0±1.2 | 96.0±1.4 | 98.4±0.5 | 48.5±2.3 |
| | carrot | 87.1±1.9 | 96.9±0.5 | 92.4±1.2 | 99.1±0.1 | 99.8±0.0 | 55.3±2.9 |
| | cookie | 81.1±6.5 | 94.4±1.9 | 90.0±1.8 | 92.4±0.7 | 97.6±0.2 | 56.9±3.4 |
| | dowel | 74.3±2.2 | 92.4±1.3 | 90.6±0.5 | 94.8±0.4 | 98.8±0.1 | 40.7±2.6 |
| | foam | 80.5±2.1 | 95.2±0.5 | 89.4±0.5 | 79.9±0.7 | 92.9±0.4 | 43.7±0.6 |
| | peach | 94.2±2.5 | 98.2±1.1 | 94.8±1.4 | 99.2±0.1 | 99.8±0.0 | 62.8±3.7 |
| | potato | 67.4±2.8 | 89.7±1.4 | 89.5±0.4 | 98.2±0.1 | 99.6±0.0 | 42.7±3.3 |
| | rope | 96.4±1.3 | 98.7±0.4 | 95.3±0.5 | 97.7±0.1 | 99.4±0.0 | 50.4±0.3 |
| | tire | 69.3±12.0 | 89.2±6.2 | 87.6±0.3 | 91.7±0.9 | 98.1±0.2 | 44.1±5.3 |
| | **Mean** | **82.8±2.9** | **94.9±1.1** | **91.6±0.6** | **94.8±0.3** | **98.4±0.1** | **51.3±1.7** |
| **MVTecLOCO** | breakfast_box | 78.1±2.1 | 88.3±0.9 | 78.8±1.5 | 64.4±1.9 | 77.7±1.5 | 29.9±1.9 |
| | juice_bottle | 58.8±3.7 | 79.2±1.6 | 83.9±0.8 | 70.9±0.7 | 85.6±0.4 | 29.4±1.0 |
| | pushpins | 57.0±2.1 | 63.3±4.4 | 72.0±0.5 | 51.8±0.8 | 53.9±1.2 | 3.0±0.3 |
| | screw_bag | 60.0±2.7 | 76.2±2.0 | 78.2±0.0 | 69.2±0.6 | 65.2±1.3 | 10.9±0.1 |
| | splicing_connectors | 69.4±4.8 | 80.2±5.0 | 77.2±0.0 | 75.0±0.7 | 64.3±2.1 | 19.3±0.8 |
| | **Mean** | **64.6±1.4** | **77.4±1.7** | **78.0±0.2** | **66.3±0.4** | **69.3±0.4** | **18.5±0.5** |
| **BTAD** | 01 | 86.4±0.2 | 95.0±0.1 | 85.9±0.6 | 65.6±2.7 | 93.5±1.1 | 52.1±1.3 |
| | 02 | 93.1±0.7 | 98.9±0.1 | 95.6±0.2 | 65.5±2.0 | 96.9±0.1 | 65.2±1.6 |
| | 03 | 99.7±0.1 | 95.5±1.8 | 91.8±2.8 | 97.2±0.0 | 99.5±0.0 | 64.6±0.5 |
| | **Mean** | **93.0±0.3** | **96.5±0.5** | **91.1±0.9** | **76.1±1.4** | **96.7±0.4** | **60.6±1.0** |
| **MPDD** | bracket_black | 64.6±8.6 | 73.6±7.7 | 77.8±2.4 | 94.0±0.9 | 96.5±0.6 | 26.5±3.8 |
| | bracket_brown | 60.8±1.4 | 75.6±1.6 | 82.0±1.0 | 89.0±0.8 | 95.4±0.5 | 13.1±1.1 |
| | bracket_white | 81.1±6.7 | 77.1±10.9 | 84.7±1.0 | 96.5±1.3 | 99.8±0.1 | 23.3±4.5 |
| | connector | 87.8±4.0 | 72.6±6.3 | 77.8±2.1 | 92.0±1.6 | 97.6±0.5 | 30.7±4.5 |
| | metal_plate | 99.9±0.1 | 100.0±0.0 | 99.5±0.4 | 90.0±0.7 | 96.6±0.4 | 73.5±1.6 |
| | tubes | 94.0±0.6 | 97.6±0.3 | 91.8±0.8 | 95.6±0.5 | 98.8±0.1 | 64.9±0.8 |
| | **Mean** | **81.4±2.9** | **82.7±3.0** | **85.6±0.3** | **92.9±0.5** | **97.5±0.1** | **38.7±1.9** |

Table 11: The comprehensive results of our NAGL framework, evaluated under settings with $N^4 + A^1$.

| Dataset | Objects | Image-level | | | Pixel-level | | |
| --- | --- | --- | --- | --- | --- | --- | --- |
| | | AUROC | AP | F1-max | AUROC | PRO | F1-max |
| MVTecAD | bottle | 99.8±0.2 | 99.9±0.1 | 99.2±0.0 | 98.1±0.2 | 95.7±0.2 | 76.6±0.7 |
| | cable | 94.7±0.7 | 97.2±0.3 | 91.2±0.5 | 95.2±0.3 | 89.4±0.8 | 61.0±1.2 |
| | capsule | 97.4±1.0 | 99.5±0.2 | 97.3±0.8 | 98.4±0.0 | 97.3±0.1 | 52.4±0.4 |
| | carpet | 100.0±0.0 | 100.0±0.0 | 100.0±0.0 | 99.3±0.0 | 98.4±0.0 | 68.5±0.2 |
| | grid | 99.8±0.3 | 99.9±0.1 | 99.7±0.5 | 99.4±0.0 | 97.0±0.1 | 51.3±0.9 |
| | hazelnut | 99.8±0.2 | 99.9±0.1 | 99.3±0.7 | 99.6±0.1 | 97.5±0.7 | 79.7±1.5 |
| | leather | 100.0±0.0 | 100.0±0.0 | 100.0±0.0 | 99.1±0.0 | 98.2±0.3 | 41.0±1.1 |
| | metal_nut | 99.8±0.0 | 100.0±0.0 | 99.1±0.3 | 96.0±0.3 | 94.0±0.1 | 72.6±1.1 |
| | pill | 95.3±1.4 | 99.1±0.3 | 96.3±0.5 | 95.3±0.2 | 96.2±0.2 | 55.0±0.9 |
| | screw | 84.4±4.2 | 94.5±1.8 | 88.4±0.9 | 98.3±0.3 | 93.8±0.7 | 47.0±1.9 |
| | tile | 100.0±0.0 | 100.0±0.0 | 99.6±0.3 | 96.0±0.3 | 91.2±0.2 | 69.5±0.5 |
| | toothbrush | 99.8±0.2 | 99.9±0.1 | 98.9±0.9 | 99.2±0.0 | 95.5±0.5 | 64.7±1.4 |
| | transistor | 85.7±5.9 | 79.9±7.0 | 79.0±4.4 | 85.8±1.2 | 67.8±1.7 | 38.4±0.9 |
| | wood | 99.8±0.3 | 99.9±0.1 | 98.9±0.5 | 96.4±0.3 | 95.7±0.3 | 69.1±1.1 |
| | zipper | 99.5±0.2 | 99.9±0.1 | 98.7±0.4 | 98.3±0.4 | 94.2±0.5 | 54.7±0.6 |
| | **Mean** | **97.1±0.6** | **98.0±0.5** | **96.4±0.2** | **97.0±0.0** | **93.5±0.2** | **60.1±0.1** |
| VisA | candle | 94.0±0.4 | 94.4±0.6 | 87.5±0.7 | 99.1±0.1 | 96.8±0.1 | 38.2±0.6 |
| | capsules | 97.5±0.2 | 98.4±0.1 | 94.5±0.6 | 98.1±0.4 | 96.8±0.5 | 48.6±4.6 |
| | cashew | 92.7±1.5 | 96.7±0.6 | 89.6±1.8 | 98.7±0.2 | 98.0±0.5 | 59.1±0.2 |
| | chewinggum | 97.6±0.1 | 99.0±0.0 | 95.5±0.3 | 99.2±0.1 | 85.7±1.2 | 68.1±0.8 |
| | fryum | 95.8±0.9 | 98.2±0.4 | 94.0±1.2 | 96.2±0.6 | 90.8±0.6 | 41.7±0.7 |
| | macaroni1 | 89.5±2.1 | 88.8±1.9 | 83.5±0.9 | 99.4±0.1 | 93.6±0.5 | 26.5±1.2 |
| | macaroni2 | 66.2±5.0 | 63.2±4.8 | 70.0±1.1 | 98.5±0.1 | 87.6±0.2 | 22.3±1.8 |
| | pcb1 | 95.4±0.7 | 93.3±0.8 | 92.7±0.8 | 98.9±0.1 | 94.2±0.1 | 60.4±2.0 |
| | pcb2 | 88.4±1.1 | 85.8±0.8 | 82.9±0.6 | 95.7±0.1 | 86.5±0.2 | 38.5±1.1 |
| | pcb3 | 89.0±1.4 | 91.6±1.1 | 83.1±2.0 | 95.2±0.5 | 87.4±0.3 | 42.5±3.9 |
| | pcb4 | 91.0±3.5 | 91.9±2.7 | 86.1±4.0 | 95.3±0.3 | 83.1±1.3 | 30.3±1.0 |
| | pipe_fryum | 97.1±1.2 | 98.6±0.6 | 93.7±1.3 | 98.9±0.1 | 97.7±0.0 | 51.3±1.0 |
| | **Mean** | **91.2±1.1** | **91.7±0.8** | **87.8±0.9** | **97.8±0.0** | **91.5±0.2** | **44.0±0.6** |
| **BraTS** | **brain** | **84.9±1.7** | **97.3±0.4** | **94.1±0.1** | **97.1±0.4** | **80.7±0.5** | **54.7±2.8** |
| MVTec3D | bagel | 96.0±1.9 | 99.0±0.5 | 95.8±1.4 | 99.0±0.1 | 99.7±0.0 | 67.0±3.2 |
| | cable_gland | 91.5±0.7 | 97.9±0.1 | 92.9±1.5 | 98.1±0.2 | 99.2±0.1 | 56.3±0.4 |
| | carrot | 90.1±1.1 | 97.7±0.3 | 93.8±0.4 | 99.3±0.1 | 99.8±0.0 | 57.7±0.5 |
| | cookie | 85.3±2.1 | 95.7±0.6 | 89.9±0.1 | 93.0±0.1 | 97.7±0.0 | 58.8±1.1 |
| | dowel | 83.4±4.9 | 95.4±2.1 | 90.3±1.0 | 95.8±0.6 | 99.1±0.2 | 46.0±6.4 |
| | foam | 79.9±5.0 | 95.1±1.2 | 90.1±1.0 | 79.7±0.7 | 90.2±0.3 | 43.9±1.4 |
| | peach | 96.8±1.5 | 99.2±0.3 | 96.3±1.2 | 99.4±0.0 | 99.8±0.0 | 69.0±0.6 |
| | potato | 68.7±3.2 | 90.9±0.7 | 89.6±0.2 | 98.2±0.2 | 99.6±0.0 | 43.9±1.7 |
| | rope | 96.8±1.2 | 98.8±0.5 | 95.6±1.3 | 97.7±0.1 | 99.4±0.0 | 50.8±0.4 |
| | tire | 80.4±4.8 | 94.2±1.3 | 88.6±1.3 | 93.2±0.2 | 98.5±0.1 | 48.5±0.4 |
| | **Mean** | **86.9±1.8** | **96.4±0.5** | **92.3±0.7** | **95.4±0.0** | **98.6±0.0** | **54.2±0.4** |
| MVTecLOCO | breakfast_box | 79.5±2.2 | 89.1±1.0 | 79.5±1.7 | 64.9±1.1 | 77.7±1.0 | 30.6±0.8 |
| | juice_bottle | 75.0±7.4 | 88.5±5.4 | 84.1±0.6 | 70.2±0.2 | 85.2±0.2 | 28.3±0.6 |
| | pushpins | 65.0±8.2 | 69.1±5.8 | 74.5±3.0 | 53.6±1.1 | 53.5±2.0 | 3.4±0.3 |
| | screw_bag | 65.5±5.3 | 79.3±2.5 | 79.0±1.4 | 70.0±0.9 | 64.7±0.2 | 10.9±0.1 |
| | splicing_connectors | 72.7±9.1 | 81.6±8.0 | 78.5±1.7 | 76.1±1.4 | 65.2±4.0 | 19.9±1.3 |
| | **Mean** | **71.5±3.1** | **81.5±2.2** | **79.1±0.8** | **66.9±0.6** | **69.2±0.5** | **18.6±0.4** |
| BTAD | 01 | 90.2±2.7 | 96.4±0.9 | 89.0±1.7 | 67.8±0.8 | 93.8±0.7 | 52.9±0.9 |
| | 02 | 93.4±0.9 | 98.9±0.2 | 95.7±0.2 | 64.3±1.7 | 96.9±0.1 | 64.9±1.6 |
| | 03 | 99.7±0.1 | 96.3±1.0 | 92.7±2.2 | 97.2±0.2 | 99.5±0.0 | 65.7±0.9 |
| | **Mean** | **94.4±1.2** | **97.2±0.5** | **92.5±0.9** | **76.4±0.9** | **96.8±0.2** | **61.2±0.4** |
| MPDD | bracket_black | 68.9±11.5 | 76.0±10.0 | 78.9±2.2 | 95.4±1.6 | 97.6±1.2 | 27.2±3.0 |
| | bracket_brown | 60.7±8.8 | 73.3±5.4 | 83.2±1.1 | 90.4±1.7 | 96.0±0.5 | 15.3±2.0 |
| | bracket_white | 72.6±17.4 | 69.7±20.9 | 82.2±6.2 | 96.5±1.4 | 99.8±0.1 | 28.2±3.6 |
| | connector | 86.3±2.7 | 74.6±3.5 | 77.2±3.9 | 92.2±1.1 | 97.7±0.3 | 36.2±6.0 |
| | metal_plate | 100.0±0.0 | 100.0±0.0 | 99.5±0.4 | 90.4±0.5 | 96.7±0.4 | 73.0±1.9 |
| | tubes | 95.8±0.8 | 98.3±0.3 | 93.8±1.1 | 96.1±0.3 | 99.0±0.1 | 66.1±0.1 |
| | **Mean** | **80.7±3.0** | **82.0±3.9** | **85.8±1.3** | **93.5±0.7** | **97.8±0.3** | **41.0±1.1** |

