# OpenReview forum: "Normal-Abnormal Guided Generalist  Anomaly Detection"
_NeurIPS.cc/2025/Conference — NeurIPS 2025 poster_

### Official Review · Reviewer_QNYq · 2025-06-20

**Clarity:** 3
**Significance:** 3
**Originality:** 3
**Rating:** 5
**Confidence:** 5

**Summary:**

This paper aims to tackle generalist anomaly detection (GAD). Previous GAD methods primarily use only normal samples as references, overlooking the valuable information in anomalous samples. To address this, this paper proposes a more practical GAD method by innovatively incorporating few -shot anomalous samples as references to guide anomaly detection across diverse domains. The proposed Normal-Abnormal Generalist Learning framework consists of two parts: Residual Mining and Anomaly Feature Learning. The method can effectively utilize both normal and anomalous references for more accurate and practical cross-domain anomaly detection.

**Questions:**

Please see Strengths And Weaknesses.

**Ethical Concerns:**

["NO or VERY MINOR ethics concerns only"]

**Final Justification:**

The authors' rebuttal can address my main concerns (please see my response to the authors), thus I decide to raise the final score for the apper.

**Limitations:**

yes

**Quality:**

3

**Strengths And Weaknesses:**

**[Strengths]**

1. This paper focuses on a previously overlooked issue in GAD method: only considering to use few-shot normal samples as references without considering to effectively utilize information from few-shot abnormal samples. This is interesting and very valuable for practical usage. This paper proposes a good GAD method that can also effectively utilize abnormal samples when directly used in unseen product classes.

2. The proposed method is simple but effective. The main innovative module in this paper is the RM-AFL module, the whole module is simply based on cross-attention to capture normal-abnormal differences in residual space. The finally learned anomaly proxies can represent the anomalous patterns in the query image, thus can highlight anomalous regions by further feature comparison.

**[Major Weaknesses]**

1. I think the anomaly classification in the title is not accurate because the model in the paper cannot generate classification labels for anomalous regions. Anomaly classification requires further classification of anomaly types based on detection, and the method in the paper only implements anomaly detection.

2. In Introduction, the first point in contribution conclusion is not reasonable. “We propose a new generalist anomaly detection task and corresponding dataset”. Generalist anomaly detection has been proposed in previous papers, and this paper is only a further study. This paper still uses the existing datasets, only divides the data into new settings, and does not propose a new dataset. The main difference between this paper and previous papers is the consideration of using both normal and abnormal samples as references.

3. Although the proposed method is simple and easy to follow, I think one main issue about the paper is that the method lacks corresponding explanations. Why the key and value in the first cross-attention are $\mathcal{F}^a$ and $Res(\mathcal{F}^a, \mathcal{F}^n)$? Why the key and value in the second cross-attention are $Res(\mathcal{F}^q, \mathcal{F}^n)$ and $\mathcal{F}^q$? Is this order necessary? What would happen if the order is swapped or different key and value are used? What do residual proxies and anomaly proxies represent? Do residual proxies represent abnormal patterns of reference abnormal samples in residual space? Do anomaly proxies represent abnormal patterns of the query sample in vision space? Why do we need attention in the residual space first? Can't we directly extract anomaly proxies from $\mathcal{F}^q$ in the vision space?

4. In Eq(5), the $\mathcal{M}^{\prime}$ is based ground-truth mask $\mathcal{M}^a$. However, the ground-truth mask during testing is not known in advance. How is the Residual Mining part implemented during testing (how is the $\mathcal{M}^\prime$ part handled)?

5. About implementation details. Is the RM-AFL module implemented with one layer? Or the RM-AFL module is implemented by stacking multiple layers. This critical detail is missing.

6. About experiments. Why does PromptAD only appear in $N^1$ setting? Why only ResAD is reproduced? Is the ResAD reproduced based on the backbone (ViT-S) and 448x448 resolution used in your method (From appendix D, it seems that ResAD is reproduced based on wide-resnet50)? If for fair comparison, other methods should also be reproduced based on ViT-S and 448x448 resolution. You may need to reproduce ResAD based on the backbone used in your method and 448x448 resolution for more reasonable comparison. Also in Tab.2, the results of other methods are from ResAD paper, but why does ResAD need to be specifically reproduced. Additionally, the experiments in main text are not sufficient, you should conduct more cross-dataset experiments to validate your method (I see experiment results on MVTec3D, MVTecLOCO, BTAD, MPDD, it’s better to reorganize your paper to include these experiment comparison in main text).

7. Strictly, your method does not just use an abnormal sample, as you randomly select a sample from each anomaly type. For a product class, there may be many anomaly types (e.g. Mvtec dataset), and the actual number of abnormal samples used may be even more than the normal samples. The more critical issue is that in practical use, we do not know the anomaly type in the query sample. In this case, how should we choose abnormal samples and which anomaly type should we choose from. You need to discuss how much influence it would have on the results if the anomaly type in the abnormal reference sample is different from the type in the query sample. And what influence does it have if multiple different anomaly types of abnormal reference samples are given to a query sample.

**[Minor Weaknesses]**

Typos: In line 118, $r^a$ misses the subscript k. In line 145, the shape of $\mathcal{M}^a$ should be $\mathbb{R}^{K_2L}$.

I think the good explanations for Weakness 3 and influence about anomaly type in Weakness 7 are important. Weakness 4 and 6 also should be well addressed. Overall, I think this paper considers an interesting and very valuable task, and proposes a simple but effective method. The entire paper is easy to follow. However, due to the aforementioned weaknesses, I am currently unable to provide an accept. If my concerns can be effectively addressed, I am willing to raise my score.

---

> ### Author Rebuttal · Authors · 2025-07-29
>
> **Dear QNYq,**
>
> Thank you for reviewing our paper. We address your questions individually below. Please let us know if you have further questions. We will answer them promptly.
>
> ***Q1. About the title.***
> > **A1.** "Anomaly Classification and Segmentation" in the title follows from [19], titled "WinCLIP: Zero-/Few-Shot Anomaly Classification and Segmentation." In their introduction, they define image-level abnormality as classification. For a more precise expression, we are considering changing the title to "Normal-Abnormal Guided Generalist Anomaly Detection" *(if the rules allow)*.
>
> ***Q2. About introduction.***
> > **A2.** We will change it to "We introduce a new task that utilizes a mixture of normal and anomalous samples as references for generalizable anomaly detection and have constructed benchmarks based on existing datasets."
>
> ***Q3. The explanations of two attentions in RM-AFL module.***
>
> > **A3.**  Yes, your understanding is correct, residual proxies represent abnormal patterns of reference abnormal samples in residual space, and anomaly proxies represent abnormal patterns of the query sample in vision space.  As validated in **Appendix B**, residual features exhibit a common distribution, even among residual features of different anomalies  (**Figure 5**). Residual features of known anomalies can provide references; if the residual feature of a query is similar to them, the corresponding visual feature is likely to be an anomaly.
>
> > In the first attention (RM), we set the value as $\text{Res}(\mathcal{F}^a,\mathcal{F}^n)$, which provides the "normal-abnormal differences" in the residual space.  We define the key as $\mathcal{F}^a$, which are the original abnormal patterns. Since the query is **learnable**, the output is the **optimally** aggregated residual features (termed as residual proxies $\tilde{\mathcal{P}}$).
>
> > The second attention (AFL) aims to learn query-related abnormal patterns by comparing "normal-abnormal differences" and "normal-query differences". We set the query as $\tilde{\mathcal{P}}$ and the key as $\text{Res}(\mathcal{F}^q,\mathcal{F}^n)$. This "comparison" is achieved by computing the attention map between $\tilde{\mathcal{P}}$ and $\text{Res}(\mathcal{F}^q,\mathcal{F}^n)$. Then the query features $\mathcal{F}^q$ are aggregated by the attention map to obtain the query-related abnormal patterns (termed as anomaly proxies $\hat{\mathcal{P}}$).
>
> > Based on the above analysis, we need to ensure that the **Value** of RM and the **Key** of AFL reside within the same space. Accordingly, we have included the experimental results as presented in the **Table R9**.
> >
> > **Experiment (I)** demonstrates that swapping the **Key** and **Value** in RM-AFL leads to a significant performance drop. This is because the output of RM-AFL is aggregation of  $\text{Res}(\mathcal{F}^q,\mathcal{F}^n)$ (residual features), which are different from the original query features ($\mathcal{F}^q$) when computing the score by **Equation (8)**. Similar phenomena are also observed in (II).
> >
> > **Experiment (III)** shows that using the original visual features as the **Key** and **Value** in RM-AFL yields satisfactory performance, but a gap still exists compared to (IV). This is because the transferability of the original visual space is limited (**Appendix B Figure 5**), making it difficult to effectively capture class-agnostic information for the final detection.
>
> **Table R9. Ablation study of RM-AFL design**
>
> ||RM-*Query*|RM-*Key*|RM-*Value*|AFL-*Key*|AFL-*Value*|I-AUROC|P-AUROC|
> |:-:|:-:|:-:|:-:|:-:|:-:|:-:|:-:|
> |(I)|$\mathcal{P}$|$\text{Res}(\mathcal{F}^a,\mathcal{F}^n)$|$\mathcal{F}^a$|$\mathcal{F}^q$|$\text{Res}(\mathcal{F}^q,\mathcal{F}^n)$|93.1|93.9|
> |(II)|$\mathcal{P}$|$\text{Res}(\mathcal{F}^a,\mathcal{F}^n)$|$\text{Res}(\mathcal{F}^a,\mathcal{F}^n)$|$\text{Res}(\mathcal{F}^q,\mathcal{F}^n)$|$\text{Res}(\mathcal{F}^q,\mathcal{F}^n)$|92.8|94.0|
> |(III)|$\mathcal{P}$|$\mathcal{F}^a$|$\mathcal{F}^a$|$\mathcal{F}^q$|$\mathcal{F}^q$|94.4|95.1|
> |(IV) Ours|$\mathcal{P}$|$\mathcal{F}^a$|$\text{Res}(\mathcal{F}^a,\mathcal{F}^n)$|$\text{Res}(\mathcal{F}^q,\mathcal{F}^n)$|$\mathcal{F}^q$|95.8|96.6|
>
> ***Q4. About $\mathcal{M}^a$.***
> > **A4.** $\mathcal{M}^a$ denotes the label for an abnormal reference. When an abnormal reference is obtained, its corresponding mask is also acquired. This design originated from our collaboration with the factory. They believe it would be easy to provide a very limited number of abnormal references(e.g., even just one) and their corresponding masks. Additionally, the VISION'24 Data Challenge, organized by ECCV 2024, can also access a few masks of abnormal samples, further underscoring the practical applications of this design.
>
> ***Q5. The layers of  RM-AFL module.***
> > **A5.** As discussed in ***Q3***, the output of RM-AFL is a query-related abnormal feature. The input of RM-AFL is a learnable proxy. The output cannot be used as input, making concatenating layers unreasonable. Additionally, while residual features offer cross-domain capability, the risk of overfitting remains (**Appendix F**). Therefore, the RM-AFL module is implemented with only one layer. As shown in **Table R10**, more layers lead to performance degradation.
>
> **Table R10. Comparing the number of RM-AFL modules**
>
> |RM-AFL|I-AUROC|P-AUROC|
> |:-:|:-:|:-:|
> |1|95.8|96.6|
> |2|95.7|96.3|
> |3|93.1|94.2|
>
>  ***Q6. About experiments.***
> > **A6.**  We address these concerns as follows:
> > - We will include comparison of PromptAD under $N^2$ and $N^4$ in our main text. Although PromptAD requires training a specific model for each product, our method consistently demonstrates superior performance. **Table R11-1/2** presents comparisons for $N^2$ and $N^4$.
> > - As stated in **Lines 244-245**, ResAD only reported AUROC results for 2/4-shot settings. For a more comprehensive comparison, we reproduced their results using the official ResAD code. And training ResAD is time-consuming; therefore, we initially only provided reproduction results for the CNN-based architecture in our main text.
> > - For a fair comparison, we also included a comparison with ViT-based methods in **Appendix D Table 7**, and reproduced ResAD with the DINOv2 ViT-S [41] backbone and $448\times 448$ resolution used in our method (**Table R11-1/2**). These results indicate that our method maintains an advantage when ResAD uses the same backbone and resolution. We will use these results for comparison in the main text.
> > - Due to space constraints, more comprehensive results and comparisons with MVTec3D, MVTecLOCO, BTAD, and MPDD will be included in the main text.
>
> **Table R11-1. Different Backbone Comparison (AUROC) under $N^2(+A^1)$**
>
> |Method|backbone|Total Parameters(M)|MVTecAD I|MVTecAD P|VisA I|VisA P|BraTS I|BraTS P|
> |:-:|:-:|:-:|:-:|:-:|:-:|:-:|:-:|:-:|
> |PromptAD|CLIP ViT-B|-|95.7|96.2|88.3|97.1|-|-|
> |ResAD$^\ast$|WideResNet50|59.2|87.2|94.8|86.6|96.5|66.2|91.5|
> |ResAD$^\ast$|DINOv2 ViT-S|82.4|90.3|95.3|85.7|96.2|66.4|93.5|
> |Ours|DINOv2 ViT-S|24.4|96.8|96.8|89.8|97.6|82.1|96.8|
>
> **TableR11-2.Different Backbone Comparison (AUROC) under$N^4(+A^1)$**
>
> |Method|backbone|Total Parame ters(M)|MVTecAD I|MVTecAD P|VisA I|VisA P|BraTS I|BraTS P|
> |:-:|:-:|:-:|:-:|:-:|:-:|:-:|:-:|:-:|
> |PromptAD|CLIP ViT-B|-|96.6|96.5|89.1|97.4|-|-|
> |ResAD$^\ast$|WideResNet50|59.2|90.7|95.8|89.3|96.8|74.9|94.3|
> |ResAD$^\ast$|DINOv2 ViT-S|82.4|93.5|96.5|89.0|96.5|74.8|94.1|
> |Ours|DINOv2 ViT-S|24.4|97.1|97.0|91.2|97.8|84.9|97.1
>
> _**Q7. Using different defect types.**_
> > **A7.** Our experiments, detailed in **Appendix A**, already include different defect type scenarios. The VisA dataset aggregates all anomalous samples per product without distinguishing between defect types. This can result in scenarios where the input query and the anomalous reference display different anomaly types (**Figure 4 b**, termed as inter-type). The **Table 1** also indicates that our method maintains a performance advantage over ResAD.
>
> > To facilitate a more intuitive comparison, we conducted additional experiments as detailed in **Table R12**. We test the *"hazelnut"* and *"carpet"* products from MVTecAD, using 'crack' and 'hole' as the test type, respectively. The results indicate that while the inter-type setting causes some performance degradation, the impact is minimal. Additionally, compared to the ResAD, our method demonstrates an advantage even when using different anomaly types. Therefore, our setting and method are not limited to the same anomaly type, which aligns with practical applications.
>
> **Table R12. Comparison of different defect types as abnormal references**
>
> |Method|Test Type|Abnormal Reference Type|I-AUROC|P-AUROC|
> |:-:|:-:|:-:|:-:|:-:|
> |ResAD|crack from hazelnut|-|90.2|96.7|
> |Ours(intra)|crack from hazelnut|crack|98.9|99.2|
> |Ours(inter)|crack from hazelnut|hole|98.8|99.2|
> |Ours(inter)|crack from hazelnut|holeandprint|98.8|99.3|
> |ResAD|hole from carpet|-|90.2|96.7|
> |Ours(intra)|hole from carpet|hole|100.0|99.5|
> |Ours(inter)|hole from carpet|color|99.1|98.4|
> |Ours(inter)|hole from carpet|colorandcut|99.7|99.4|
>
> > Our method, which is based on NN search, benefits from an increased number of references, as also discussed in **Appendix C**.
>
> ***Q8. Typos.***
> > **A8.** We have corrected typos in the main text and carefully reviewed all statements.

---

> > ### Comment · Reviewer_QNYq · 2025-08-01
> >
> > Thank you for your careful responses. Based on your responses, I think my main questions (weakness 3, 4, 6, 7) have been addressed. But for the modification in Q2, it is not recommended to use the "new task", because although it is different from the previous generalist anomaly detection, it is also not a new task, and the "benchmark" is also unreasonable. Benchmarks usually require proposing new datasets and conducting a large number of experiments, and you only redivide the dataset, which cannot be considered a benchmark. I think the contribution 1 in the introduction needs to be reconsidered. In Q5, you mentioned that concatenating layers is unreasonable, but you still provided results of multiple layers, and the performance doesn't significantly decrease at two layers. This actually shows that your explanation is very far-fetched (you don't need to further reply to me, this will not cause my new negative views on the paper). I hope that in the revised paper, the title can be changed to "anomaly detection", and the results and discussions about other issues mentioned above can be well included in the revised paper. Well, I'm glad to raise my score for the paper.

---

> ### Author Response · Authors · 2025-08-01
>
> Thank you for your valuable feedback and for raising the score. We will carefully incorporate the results and discussions into the revised paper.

---

### Official Review · Reviewer_RxZT · 2025-06-30

**Clarity:** 3
**Significance:** 2
**Originality:** 2
**Rating:** 4
**Confidence:** 4

**Summary:**

This paper focuses on generalist anomaly detection that can detect anomalies in new target domains. The proposed method leverages both normal and anomalous samples as references to guide anomaly detection across diverse domains. Experiments are conducted on several popular datasets.

**Questions:**

See the weakness.

**Ethical Concerns:**

["NO or VERY MINOR ethics concerns only"]

**Final Justification:**

The authors have addressed part of my concerns about the comparison with existing methods.

**Limitations:**

See the weakness.

**Quality:**

3

**Strengths And Weaknesses:**

1. The paper lacks important citations. In few-shot anomaly detection, the utilization of limited anomalous data has been extensively studied [a,b,c]. The authors should adequately cite these papers and compare their method with them. In the current experiments, none of the competing methods use anomalous data, while the proposed method requires it. This comparison is unfair.
2. The authors define an original data domain and a target domain. However, both data domains here consist of industrial defect detection data without fundamental differences. The authors should refer to MVFA [a] and adopt medical imaging data as the target domain to demonstrate the generalization capability of the proposed method.

[a] Adapting Visual-Language Models for Generalizable Anomaly Detection in Medical Images. CVPR 2024.
[b] Catching both gray and black swans: Open-set supervised anomaly detection. CVPR 2022.
[c] Explicit boundary guided semi-pushpull contrastive learning for supervised anomaly detection. CVPR 2023.

---

> ### Author Rebuttal · Authors · 2025-07-29
>
> **Dear RxZT,**
>
> Thank you for reviewing our paper. We appreciate your suggestion to strengthen our citations and comparisons, as this helps contextualize our work. We address each of your questions below and are available to answer any further questions promptly.
>
> ***Q1. Lack of citations.***
> > **A1.** We will include references to [a] in our main text, while [b, c] have already been cited (please refer to [11, 53]). Since the settings of [a, 11, 53] differ significantly from ours, it is difficult to compare our methods with theirs. Instead, our method is more similar to [52, 61]; therefore, we primarily discussed [52, 61] in the main text. Additionally, for clearer comparison, we will now detail the differences between our method and those in [a, 11, 53]:
> >
> > **Comparison with [a].** The method described in [a] is specifically designed for medical image detection. It involves transferring knowledge from one medical dataset to another, not from an industrial dataset to a medical one, as we do in **Table 2**. Additionally, to directly compare the differences, we provide some experimental results in ***Q2***.
> >
> > **Comparison with [11, 53].** The method described in [11, 53] focuses on _open-set supervised anomaly detection (OSAD)_, which has two experiment protocols, including multi-class and one-class settings. For the multi-class setting, a few labeled anomaly samples are randomly drawn from all possible anomaly classes in the test set per dataset. These sampled anomalies are used for training, and then removed from the test data. For the one-class setting, the anomaly example sampling is limited to one single anomaly class only, and all anomaly samples in this anomaly class are used for training and removed from the test set to ensure that the test set contains only unseen anomaly classes.
>
> > This means that these methods [11, 53] require training a specific model for each dataset for anomaly detection, which is different from our method. Our method applies meta-learning on the original domain to facilitate cross-domain transfer (**no training on target domains**).
>
> > **Table R7** presents a comparison of our method with methods in [11] and [53]. *(The results for [11] are obtained from Tables 1 and 2 of their original paper, while the results for [53] are obtained from Tables 1 and 4 of their original paper. And our subset-level results are reported in **Appendix G**)*. Through comparison, we found that our method performs comparably to OSAD on "carpet" data and exhibits a slight performance decrease on "metalnut" data. However, our method remains good practical applicability due to its domain-agnostic ability.
>
> **Table R7. Comparison with OSAD methods**
>
> |   Method   |       Setting       | Test Product |    Training anomaly types    | I-AUROC | P-AUROC |     Model Type      |
> | :--------: | :-----------------: | :----------: | :--------------------------: | :-----: | :-----: | :-----------------: |
> |  DRA [11]  |     multi-class     |    carpet    | random sampling from MVTecAD |  94.0   |    -    |   domain-specific   |
> |  DRA [11]  |      one-class      |    carpet    |  select one type in carpet   |  93.5   |    -    |   class-specific    |
> | BGAD [53]  |     multi-class     |    carpet    | random sampling from MVTecAD |  99.6   |  98.9   |   domain-specific   |
> | BGAD [53]  |      one-class      |    carpet    |  select one type in carpet   |  99.9   |  99.5   |   class-specific    |
> |    Ours    | general ($N^1+A^1$) |    carpet    |             VisA             |  100.0  |  99.3   | **domain-agnostic** |
> |  DRA [11]  |     multi-class     |   metalnut   | random sampling from MVTecAD |  99.7   |    -    |   domain-specific   |
> |  DRA [11]  |      one-class      |   metalnut   | select one type in metalnut  |  94.5   |    -    |   class-specific    |
> | BGAD [53]  |     multi-class     |   metalnut   | random sampling from MVTecAD |  99.6   |  97.0   |   domain-specific   |
> | BGAD [53]  |      one-class      |   metalnut   | select one type in metalnut  |  99.8   |  97.5   |   class-specific    |
> |    Ours    | general ($N^1+A^1$) |   metalnut   |             VisA             |  99.4   |  95.2   | **domain-agnostic** |
>
>
> ***Q2. Evaluation in the medical domain.***
> > **A2.** As described in **Lines 32-41**, our study employs a cross-domain setting and proposes a meta-learning approach for anomaly detection across diverse domains. We also validate the cross-domain transfer capabilities from industrial to medical applications in **Table 2** and **Figure 3**. Following the reviewer's suggestion, we compared our method with MVFA[a] on additional medical datasets (LiverCT and RESC), as shown in the **Table R8**. Results indicate our method demonstrates strong competitiveness, despite being trained on MVTecAD (industrial domain $\rightarrow$ medical domain).
>
> **Table R8. Comparison in the medical domain**
>
> |  Setting  |  Method  | BraTS (I-AUROC) | BraTS (P-AUROC) | LiverCT (I-AUROC) | LiverCT (P-AUROC) | RESC (I-AUROC) | RESC (I-AUROC) |
> | :-------: | :------: | :-------------: | :-------------: | :---------------: | :---------------: | :------------: | :------------: |
> |   $N^4$   | WinCLIP  |      67.3       |      93.2       |       67.2        |       96.8        |      88.8      |      96.7      |
> |   $N^4$   | MVFA [a] |        -        |        -        |       81.2        |       99.7        |      96.2      |      98.7      |
> | $N^4+A^1$ |   Ours   |      84.9       |      97.1       |       84.3        |       99.4        |      97.1      |      99.0      |

---

> > ### Comment · Reviewer_RxZT · 2025-08-06
> >
> > Thank you for the response.
> >
> > My main concern is that all the tables and methods compared in the current main paper do not utilize anomalous data, which is clearly unfair. In the three works [a,b,c] I listed that already used anomalies, MVFA [a] achieved generalizable zero-shot anomaly detection across data domains and even modalities. It was not designed specifically for medical data as the author stated in their rebuttal, and experimental results on industrial datasets were also provided in the original paper. Several subsequent papers also followed MVFA's benchmark, although I did not mention it. The other two works [b,c] I mentioned, although domain specific, are classic papers on anomaly detection using anomalous data. In the rebuttal, the author only provided comparison results for two out of nearly 20 industrial products (classes), and the selected products all had results approaching 100. Such comparisons are not very meaningful.
> >
> > Also, their contributions are overclaimed, for example, "a new generalist anomaly detection task" (a more generalizable setting even across data modalities is introduced in MVFA compared to this paper) and "new corresponding dataset" (the datasets used are all from other papers), "This task is the first to adopt a mixture of normal and abnormal samples as reference" (obvious NOT the first). This can easily mislead readers.

---

> > ### Comment · Reviewer_RxZT · 2025-08-07
> >
> > Thank you for your response. Please insert the above content into the main paper where it fits best. I hope the paper and our discussion will positively contribute to anomaly detection in this field. I will raise my score.

---

> ### Author Response · Authors · 2025-08-07
>
> Thank you for your valuable feedback.
>
> 1. **Fairness in Method Comparison**
> > The reviewer's statement that "methods compared in the current main paper do not utilize anomalous data" is inaccurate. Both InCTRL [61] and ResAD [52], which we compared, used both normal and abnormal samples during training. This is similar to the three papers (MVFA [a], DRA [11], and BGAD [53]) mentioned by the reviewer, all of which utilized both normal and abnormal samples in training phases. Therefore, our comparative methods in the main paper, InCTRL and ResAD, utilize anomalous data.
>
> 2. **The Differences between [11, 53], [61, 52] and [Ours]**
> > The primary difference lies in the testing phase.
> > - [11, 53] did not use samples as references during testing, and they are **not** designed for cross-domain (training and testing are conducted in different domains) detection.
> > - [61, 52] only used **normal samples as references** during testing, and are designed for cross-domain detection.
> > - [Ours] uses **both normal and abnormal samples as references** during testing, and are designed for cross-domain detection.
> >
> > Therefore, [61, 52, Ours] can all be considered few-shot anomaly detection (FSAD) methods. Furthermore, [61, 52, Ours] and our setting are all cross-domain methods and are thus referred to as generalist anomaly detection (GAD) methods.
>
> > **Based on the analyses in 1 and 2, our method is more appropriately compared with [61, 52] than with [11, 53].**
>
> 3. **The Generalization of MVFA [a] and GAD [61, 52, Ours]**
> > The generalization of MVFA is based on zero-shot learning (**without reference**). In contrast, the generalization in [61, 52, Ours] is based on few-shot meta-learning (**with reference**). These two settings are very different, so comparing them is inappropriate. Additionally, the design of [61, 52, Ours] aims to facilitate cross-domain detection while leveraging information from reference samples to enhance detection performance. For instance, the results in **Table 1** indicate that providing reference samples significantly improves the performance of cross-domain tasks.
>
> **Table 1. Comparison with ZSAD methods on the medical domain**
>
> |Setting|Method|BraTS (I-AUROC)|BraTS (P-AUROC)|LiverCT (I-AUROC)|LiverCT (P-AUROC)|RESC (I-AUROC)|RESC (I-AUROC)|
> |:-:|:-:|:-:|:-:|:-:|:-:|:-:|:-:|
> |Zero-shot|WinCLIP[19] (cross-domain)|-|-|64.2|96.2|42.5|80.6|
> |Zero-shot|MVFA[a] (cross-domain)|-|-|76.2|97.9|83.3|92.1|
> |$N^4$|WinCLIP[19] (in-domain)|67.3|93.2|67.2|96.8|88.8|96.7|
> |$N^4$|MVFA[a] (in-domain)|-|-|81.2|99.7|96.2|98.7|
> |$N^4+A^1$|Ours (cross-domain)|84.9|97.1|84.3|99.4|97.1|99.0|
>
> 4. **Evaluation of MVFA on Industrial Datasets.**
> > - While the MVTecAD dataset is evaluated in Table 3 of MVFA, this evaluation is conducted in a manner of **in-domain** (both training and testing on MVTecAD). This is different from [61, 52, Ours] (training on VisA and test on MVTecAD).
> > - Though not a fair comparison, as it is requested by the reviewer, we conduct a comparison in **Table 2**, indicating our method retains advantages even without training on MVTecAD.
>
> **Table 2. Evaluation on MVTecAD**
>
> |Method(referencenumber)|I-AUROC|P-AUROC|type|
> |:-|:-:|:-:|:-:|
> |[r1] AnomalyCLIP (zero-shot)|91.5|91.1|**cross-domain**|
> |[19] WinCLIP (zero-shot)|91.8|85.1|**cross-domain**|
> |[19] WinCLIP ($N^4$)|95.2|96.3|in-domain|
> |[a] MVFA ($N^4$)|96.2|96.3|in-domain|
> |[52] ResAD ($N^4$)|90.7|95.8|**cross-domain**|
> |Ours ($N^4+A^1$)|97.1|97.0|**cross-domain**|
>
> *[r1] AnomalyCLIP: Object-agnostic Prompt Learning for Zero-shot Anomaly Detection. ICLR 2024*

---

> ### Author Response · Authors · 2025-08-07
>
> 5. **Why carpet and metalnut are compared in Table R7**
> > - The methods [11, 53] referenced by the reviewer include two settings (multi-class and one-class, details  in our response to **RxZT** (**A1**)). However, only the performance for "carpet" and "metalnut" is reported across both settings in MVTecAD. For example, the Table 4 of [53] only reports the performance of "carpet" and "metalnut". Therefore, our comparisons are based on papers provided by the reviewer, rather than a selection to favor our method.
> > - Additionally, in our response to **RxZT** (**A1**), we clarified that our subset-level results are presented in **Appendix G**, which allows for easy comparison of the performance of various products, not just limited to  "carpet" and "metalnut".
>
> 6. **The description of the first contribution**
> > We are happy to change it to "We introduce a different task that utilizes a mixture of normal and anomalous samples as references for generalizable anomaly detection". The reviewer said: "This task is the first to adopt a mixture of normal and abnormal samples as reference (obvious NOT the first)." Please provide a work that utilizes "a mixture of normal and abnormal samples as **reference** in cross-domain scenarios", so we can compare with it.
>
> > In summary, we are happy to compare with the MVFA method in our main text. However, we suggest that the reviewer carefully review works [r1, 52, 61] to gain a clearer understanding of the reference-based generalist anomaly detection task.  We are available to provide further clarification and address any additional questions.
>
> *[r1] AnomalyCLIP: Object-agnostic Prompt Learning for Zero-shot Anomaly Detection. ICLR 2024*

---

> ### Author Response · Authors · 2025-08-07
>
> Thank you for your valuable feedback and for raising the score. We will carefully incorporate the results and discussions into the revised paper.

---

### Official Review · Reviewer_QEzS · 2025-07-01

**Clarity:** 2
**Significance:** 2
**Originality:** 3
**Rating:** 4
**Confidence:** 4

**Summary:**

This paper addresses the problem of Generalist Anomaly Detection (GAD), detecting anomalies in new target domains with models trained on a source domain. To this end, the authors propose a new framework, NAGL, including two key components, Residual Mining (RM) and Anomaly Feature Learning (AFL). Given an abnormal reference, RM obtains residuals between normal and abnormal samples to learn abnormal patterns. AFL learns abnormal features of a query image based on the learned abnormal patterns. NAGL compares abnormal patterns between the query and reference images to detect anomalies. The authors demonstrate the effectiveness of NAGL on standard benchmarks, and show an expensive analysis.

**Questions:**

Please refer to the strengths and weaknesses section for the details.

**Ethical Concerns:**

["NO or VERY MINOR ethics concerns only"]

**Final Justification:**

Thank you for the rebuttal. The major concerns, regarding abnormal references, experimental settings, and the effectiveness of the methods, have been sufficiently addressed. Considering both the questions raised by other reviewers and your response, I will raise the rating accordingly. I have no follow-up questions.

**Limitations:**

yes

**Quality:**

3

**Strengths And Weaknesses:**

**Strengths**

(+) NAGL incorporates abnormal references along with the normal ones, enabling effective anomaly detection across diverse domains.

(+) The proposed method is highly practical due to its lightweight design and achieves faster training and inference than the previous works [51, 62].

(+) The proposed method shows strong cross-domain generalization performance (e.g., MVTecAD → VisA, MVTecAD → BraTS).


**Weaknesses**

(-) The proposed method relies heavily on abnormal references. I am thinking that this might limit the applicability in certain real-world scenarios, where only normal data is available, making it less practical than the works of [51, 62].

(-) The authors set 𝐾_1 ≥ 𝐾_2 assuming a real-world scenario where abnormal samples are scarce. The inference setting on MVTecAD, however, seems to contradict the assumption. Specifically, if K_1=1 and K_2=1, the number of normal and abnormal references are 15 and 73, respectively. Please clarify this.

(-) The authors demonstrate that the proposed method generalizes well in cross-domain settings, such as the MVTecAD ⟶ BraTS scenario. However, the source domain is limited to MVTecAD, which contains a variety of abnormal references. It would be helpful if the authors could provide more results using other source domains.

(-) Please clarify the experimental settings (iii) in Table 3. I am wondering why (iii) shows inferior performances than (i) although (iii) exploits abnormal samples.

(-) The author shows in Table 3 the effectiveness of the proposed method that leverages abnormal references for GAD. However, it seems that NAGL adopts different backbone ones from [52, 61]. It would be more convincing if the authors provide additional results using the same backbone in [52, 61]. This would help to establish the validity of the proposed method more convincingly.

(-) There are some typos.
L82: augmentation-based -> and augmentation-based
L148: pervious -> previous
Caption of Figure1: cause -> causes
Caption of Table 3: represent -> represents, merge -> merges

---

> ### Author Rebuttal · Authors · 2025-07-29
>
> **Dear QEzS,**
>
> Thank you for reviewing our paper. Your insightful suggestions significantly improved its clarity. We address your questions individually below and are available to clarify any further questions you may have.
>
> ***Q1. The applicability of using anomalous samples in real-world scenarios.***
> > **A1.** Our task significantly differs from [51, 62]. Therefore, we guess the reviewer is interested in a discussion comparing the practicality of ours with [52, 61]. While current mainstream research [52, 61] primarily focuses on scenarios where only normal data is available, our collaborations with factories reveal that the presence of a small number of anomalous samples is often more common. This is because, as production progresses, some anomalous samples or new anomaly types can always be collected. Additionally, the VISION'24 Data Challenge, organized by ECCV 2024, focuses on industrial defect segmentation using a limited number of anomalous samples (more details are available on their official website). Over 80 teams worldwide participated, underscoring the practical applicability of addressing industrial defects with minimal anomalous data.
> > Driven by real-world industrial needs, we propose this task that utilizes a limited number (e.g., even just one) of anomalous samples. Our method also provides preliminary evidence that a small number of anomalous samples hold significant potential for achieving the goal of both speed and accuracy. Reviewer QNYq positively evaluated this problem, stating that "this paper considers an interesting and very valuable task." Therefore, we believe this setting has both research and application value.
>
> ***Q2. The number of normal and abnormal references.***
> > **A2.** It should be noted that the MVTecAD dataset distinguishes between different anomaly types, containing $15$ products and $73$ anomaly types. When $K_1=1$ and $K_2=1$, based on the product category and anomaly type of the query sample to be tested, we select $1$ normal sample of the corresponding product as a normal reference and $1$ abnormal sample of the corresponding anomaly type as an abnormal reference. The numbers $15$ and $73$ here indicate that there are a total of $15$ normal references and $73$ abnormal references for the MVTecAD dataset, but it does not mean that we use all references for each query sample. Additionally, for the VisA dataset, which does not distinguish anomaly types, we select $1$ normal reference and $1$ abnormal reference for each product. Therefore, there are $12$ normal references and $12$ abnormal references in the VisA dataset. This strategy for selecting normal references is consistent with existing few-normal-shot AD methods (e.g., PromptAD [25], InCTRL [61], and ResAD [52]).
>
> ***Q3. Evaluation on different training datasets.***
> > **A3.** Training on MVTeAD and testing on datasets such as VisA and BraTS, is to align with previous works [r1, 52, 61], facilitating comparison and analysis with these existing studies. Furthermore, to further validate the cross-domain capabilities, we conducted additional experiments, as shown in the **Table R5**. In this experiment, we used VisA and BTAD as training sets. These datasets contain less diversity with fewer products and do not distinguish between anomaly types. Specifically, VisA has $12$ products ($962$ normal samples and $1200$ abnormal samples), while BTAD has only $3$ products ($176$ normal samples and $282$ abnormal samples).
>
> > **Table R5** shows that training on VisA did not result in significant performance degradation. However, performance degradation is observed when training on BTAD for both our method and ResAD. This is likely due to the limited data volume in BTAD, which may hinder the models from adequately capturing the "differences" between normal and anomalous samples. It is noteworthy that our method experienced a smaller performance decrease compared to ResAD. And we also report the performance on other datasets, including MVTec3D, MVTecLOCO, BTAD, and MPDD, in the **Appendix Table 8-11**, which indicates our method has good cross-domain capabilities.
>
> **Table R5. Evaluation on different training datasets.**
>
> | Method | Training Dataset | Test Dataset | I-AUROC | P-AUROC | $\Delta$  |
> | :----: | :--------------: | :----------: | :-----: | :-----: | :-------: |
> | ResAD  |     MVTecAD      |    BraTS     |  73.5   |  91.0   |           |
> | ResAD  |       BTAD       |    BraTS     |  71.8   |  89.6   |   - 3.1   |
> |  Ours  |     MVTecAD      |    BraTS     |  78.1   |  96.5   |           |
> |  Ours  |       VisA       |    BraTS     |  77.6   |  96.7   |   - 0.3   |
> |  Ours  |       BTAD       |    BraTS     |  77.2   |  95.8   | - 1.6 |
>
>
> > *[r1] AnomalyCLIP: Object-agnostic Prompt Learning for Zero-shot Anomaly Detection. ICLR 2024*
>
> ***Q4. Clarification of settings in Table 3.***
> > **A4**. We will now explain the different settings in Table 3 individually.
> >
> > **setting (i):** As discussed in Section 4.3, we apply a nearest neighbor (NN) search between query features ($\mathcal{F}^q$) and normal reference features ($\mathcal{F}^n$), as detailed in **Equations (8,9)**.
> >
> > **setting (ii):** Similar to setting (i), we also apply the NN search between query features ($\mathcal{F}^q$) and abnormal reference features ($\hat{\mathcal{F}}^{a}$). It is important to note that $\hat{\mathcal{F}}^{a}$ is a subset of $\mathcal{F}^{a}$,  and exclusively comprises abnormal patch features from $\mathcal{F}^{a}$. This process yields an additional anomaly score map derived from the abnormal reference, which can be formulated as:
> >
> > $f^a_* :=\mathop{\mathbf{argmin}}\_{f^{a} \in \hat{\mathcal{F}}^{a}}\left(\mathbf{d}(f\_i^{q},f^{a})\right),$
> >
> > $\mathcal{S}^{i}_{a}=1-\mathbf{d}(f^q_i, f^a _*).$
> >
> > **setting (iii)** is the result of merging settings (i) and (ii), such that:
> >
> > $\mathcal{S}^i\_{\text{(iii)}} = \frac{1}{2}(\mathcal{S}^i_{n}+\mathcal{S}^i_{a}).$
> >
> > Setting (ii) indicates that simply using a Nearest Neighbor (NN) search between query and abnormal features leads to severe over-activation (visualization results can be found in Figure 1(c)). This over-activation causes setting (iii) to perform worse than setting (i).
>
> ***Q5. Verify the effectiveness of the proposed method more convincingly.***
> > **A5.** *(It should be noted that the results in Table 3 do not use different backbones; all are based on the ViT-based [41] backbone. Next, we will primarily discuss the backbones in Tables 1 and 2.)* As described in Lines 244-245, ResAD[52] only reported results for 2/4-shot settings using AUROC metrics. To provide a more comprehensive comparison, we reproduced their results using the official ResAD code. Training ResAD is time-consuming; for instance, as shown in **Table 4**, the CNN-based backbone alone requires 20 hours of training. Training with the official ViT-based backbone was estimated to take more time. Therefore, to meet the submission deadline, we initially only provided reproduction results for the CNN-based architecture. For a fair comparison, we also included a comparison with ViT-based methods in **Appendix D**, with results presented in **Table 7**. Furthermore, for a more rigorous comparison, we also reproduced ResAD with the DINOv2 ViT-S [41] backbone used in our method, and the results are shown in **Table R6-1/2**. *(Since InCTRL[61] uses text prompts, implying its reliance on a CLIP-based backbone, we maintained the original InCTRL backbone when reporting our results.)*  **Table R6-1/2** indicates that our method maintains its advantage.
>
> **Table R6-1. Different Backbone Comparison under $N^2(+A^1)$**
>
> |Method|backbone|Total Parameters (M)|MVTecAD I-AUROC|MVTecAD P-AUROC|VisA I-AUROC|VisA P-AUROC|BraTS I-AUROC|BraTS P-AUROC|
> |:-:|:-:|:-:|:-:|:-:|:-:|:-:|:-:|:-:|
> |InCTRL|CLIP ViT-B|117.5|94.0|-|85.8|-|74.6|-|
> |ResAD|Imagebined_huge|442.6|94.4|95.6|84.5|95.1|67.9|94.3|
> |ResAD$^\ast$|WideResNet50|59.2|87.2|94.8|86.6|96.5|66.2|91.5|
> |ResAD$^\ast$|DINOv2 ViT-S|82.4|90.3|95.3|85.7|96.2|66.4|93.5|
> |Ours|DINOv2 ViT-S|24.4|96.8|96.8|89.8|97.6|82.1|96.8|
>
> **TableR6-2.Different Backbone Comparison under $N^4(+A^1)$**
>
> |Method|backbone|Total Parameters (M)|MVTecAD I-AUROC|MVTecAD P-AUROC|VisA I-AUROC|VisA P-AUROC|BraTS I-AUROC|BraTS P-AUROC|
> |:-:|:-:|:-:|:-:|:-:|:-:|:-:|:-:|:-:|
> |InCTRL|CLIP ViT-B|117.5|94.5|-|87.7|-|76.9|-|
> |ResAD|Imagebined_huge|442.6|94.2|96.9|90.8|97.5|84.6|96.1|
> |ResAD$^\ast$|WideResNet50|59.2|90.7|95.8|89.3|96.8|74.9|94.3|
> |ResAD$^\ast$|DINOv2 ViT-S|82.4|93.5|96.5|89.0|96.5|74.8|94.1|
> |Ours|DINOv2 ViT-S|24.4|97.1|97.0|91.2|97.8|84.9|97.1|
>
> ***Q6. Some typos***
> > **A6.** We have corrected typos in the main text and carefully reviewed all statements.

---

> > ### Comment · Reviewer_QEzS · 2025-08-07
> >
> > Thank you for the rebuttal. The major concerns, regarding abnormal references, experimental settings, and the effectiveness of the methods, have been sufficiently addressed. Considering both the questions raised by other reviewers and your response, I will raise the rating accordingly. I have no follow-up questions.

---

> ### Author Response · Authors · 2025-08-07
>
> Thank you for your valuable feedback and for the increased score.

---

### Official Review · Reviewer_LrM6 · 2025-07-03

**Clarity:** 3
**Significance:** 3
**Originality:** 3
**Rating:** 4
**Confidence:** 5

**Summary:**

This paper proposes a novel generalist anomaly detection (GDA) framework, termed Normal-Abnormal Generalist Learning (NAGL), which leverages both normal and abnormal reference samples during training and inference to reflect more realistic application scenarios where limited anomaly examples may be available. The framework integrates two key components: Residual Mining (RM), which extracts transferable anomaly representations from the residuals between normal and abnormal references, and Anomaly Feature Learning (AFL), which identifies discriminative anomaly features in query images by comparing them against these residual proxies. Designed for cross-domain generalization, NAGL achieves strong performance across diverse datasets such as MVTecAD, VisA, and BraTS. Extensive experiments, including ablation studies and efficiency analysis, demonstrate its effectiveness and computational efficiency, highlighting its potential for practical anomaly detection applications.

**Questions:**

1. How does the performance vary when abnormal references are from different defect types than the query (e.g., inter-type transfer)? Is there any observed instability or degradation?

2. While BraTS provides a cross-domain test, have you evaluated zero-shot performance on unseen object classes in the same dataset? This would help assess the method’s generalization depth.

3. AFL uses residual attention to avoid false activations. How does this behave in cluttered scenes or images with complex textures? Would AFL overfit to known anomaly shapes?

**Ethical Concerns:**

["NO or VERY MINOR ethics concerns only"]

**Final Justification:**

After the discussion, my concerns have been addressed. As the response covers many aspects that need to be improved, these should be carefully incorporated into the final version. I keep my score as "borderline accept".

**Limitations:**

yes

**Quality:**

3

**Strengths And Weaknesses:**

Strengths:
1. The paper is the first to propose GAD with both normal and abnormal reference samples, aligning better with practical settings where limited abnormal data may exist.
2. The RM-AFL mechanism is well-motivated and shown to resolve issues of false activations when using abnormal references naively.

Weaknesses
1. The selection of abnormal references is random (per defect type), but its potential impact on performance variability is not well studied or controlled.
2. The RM and AFL modules, while empirically strong, could benefit from clearer theoretical motivation or formal insight into their robustness or transferability.
3. The method assumes access to both normal and abnormal references at inference time, which may not always hold in fully open-world applications.

---

> ### Author Rebuttal · Authors · 2025-07-29
>
> **Dear LrM6,**
>
> Thank you for reviewing our paper. Your insightful suggestions have significantly improved its clarity. We have addressed your comments one by one. Please do not hesitate to let us know if you have any further questions.
>
> _**Q1. Impact of random abnormal reference selection on performance variability.**_
> > **A1.** To clarify, our training strategy employs random sampling to construct input pairs (normal images, an abnormal image, and a query image), thereby enriching the diversity of our training samples. **For testing**, we maintain consistency by using the same abnormal reference for a given anomaly type. For instance, when evaluating the 'broken_large' anomaly type for the 'bottle' product in the MVTecAD dataset, we select the first image as the abnormal reference and apply it to test the remaining data. This approach ensures that our testing benchmark is well-controlled. We will also provide the list of reference files used during testing when we release our code. We also provide a detailed description of our dataset structure in the **Appendix A**. Please refer to "Dataset Collection" (lines 451-498) to pursue a clear understanding.
>
> > Moreover, the paper rigorously evaluates performance variability through statistical reporting: results are averaged across multiple independent runs with different random seeds (Lines 238-239), and detailed subset-level results include standard deviations (e.g., $mean \pm std$ in **Appendix, Tables 9–11**). For comparative purposes, we present the variability of different methods in **Table R1**. WinCLIP and InCTRL performance data are sourced from Tables [1, 4] and Table [1] of their original papers, respectively. The results indicate that our method is more stable, exhibiting a smaller standard deviation.
>
> **Table R1. Comparison of stability ($mean \pm std$) under $N^2(+A^1)$**
>
> |Dataset$\rightarrow$|MVTecAD|MVTecAD|VisA|VisA|
> |:-:|:-:|:-:|:-:|:-:|
> |Method$\downarrow$|I-AUROC|P-AUROC|I-AUROC|P-AUROC|
> |WinCLIP|94.4±1.3|96.0±0.3|84.6±2.4|96.8±0.3|
> |InCTRL|94.0±1.5|-|85.8±2.2|-|
> |Ours|96.8±0.8|96.8±0.2|89.8±1.5|97.6±0.0|
>
> _**Q2. Clearer theoretical motivation or formal insight for RM and AFL modules.**_
> > **A2.**  The RM and AFL are two attentions. In attention mechanisms, the Query and Key are initially used to compute the attention map. Then, this map is used to weight and combine the Value, meaning the output of the attention mechanism primarily represents the Value.
>
> > In the first attention (RM), we set the value as $\text{Res}(\mathcal{F}^a,\mathcal{F}^n)$, which provides the "normal-abnormal differences" in the residual space.  We define the key as $\mathcal{F}^a$, which are the original abnormal patterns. Since the query is **learnable**, the output is the **optimally** aggregated residual features (termed as residual proxies $\tilde{\mathcal{P}}$).
>
> > The second attention (AFL) aims to learn query-related abnormal patterns by comparing "normal-abnormal differences" and "normal-query differences". We set the query as $\tilde{\mathcal{P}}$ and the key as $\text{Res}(\mathcal{F}^q,\mathcal{F}^n)$. This "comparison" is achieved by computing the attention map between $\tilde{\mathcal{P}}$ and $\text{Res}(\mathcal{F}^q,\mathcal{F}^n)$. Then the query features $\mathcal{F}^q$ are aggregated by the attention map to obtain the query-related abnormal patterns (termed as anomaly proxies $\hat{\mathcal{P}}$).
>
> > Detailed analysis of transferability in residual space: As validated in **Appendix B**, features in the original visual space cluster by anomaly type (**Figure 5 a**), indicating strong visual correlations that impede cross-domain learning. Conversely, features in the residual space (**Figure 5 b**) do not exhibit such clustering; instead, they converge to a common distribution. This demonstrates that residual features effectively reduce type-specific information, enhancing their transferability.
>
> _**Q3. Assumption of abnormal reference access at inference time.**_
> > **A3.** As discussed in Lines 34-35, real-world scenarios often present a limited number of anomalous samples (e.g., defective parts or diagnosed disease cases), which originates from our direct collaboration with factories. More specifically, in industrial production, the scarcity of anomalous samples is common during the initial production phase. However, as production increases, anomalous samples become more readily available. The practical relevance of this challenge is further highlighted by the VISION'24 Data Challenge, organized by ECCV 2024, which focused on industrial defect segmentation with limited anomalous samples. The ability to leverage anomaly samples to enhance detection performance is highly desirable for many factories, and this paper addresses precisely this scenario. Our method also provides preliminary evidence that a small number of anomalous samples hold significant potential for achieving both speed and accuracy. Furthermore, reviewer QNYq positively evaluated this problem, stating that "this paper considers an interesting and very valuable task." Therefore, we believe this setting is valuable for research and application
>
> _**Q4. Performance when abnormal references are from different defect types.**_
> > **A4.** Our experiments already include inter-type scenarios, as detailed in **Appendix A**. The VisA dataset combines all anomaly samples for each product. This can lead to situations where the input query and the anomalous reference display different types of anomalies (**Figure 4 b**). Nevertheless, our method demonstrates an advantage over ResAD in the experimental results (**Table 1**).
>
> > For a clearer comparison, we conducted further experiments (**Table R2**) on the MVTecAD dataset. We tested "hazelnut" and "carpet" products under the $N^1+A^1$ setting, using 'crack' and 'hole' as anomaly types. Results show that while the inter-type setting causes slight degradation, our method still outperforms ResAD (which lacks abnormal references), even with different anomaly types. This aligns with our ***Q2*** discussion, indicating the transfer capacity of these residual features.
>
> **Table R2. Comparison of different defect types as abnormal references**
>
> |Method|Test Type|Abnormal Reference Type|I-AUROC|P-AUROC|
> |:-:|:-:|:-:|:-:|:-:|
> |ResAD|crack from hazelnut|-|90.2|96.7|
> |Ours(intra)|crack from hazelnut|crack|98.9|99.2|
> |Ours(inter)|crack from hazelnut|hole|98.8|99.2|
> |Ours(inter)|crack from hazelnut|print|98.6|99.0|
> |ResAD|hole from carpet|-|90.2|96.7|
> |Ours(intra)|hole from carpet|hole|100.0|99.5|
> |Ours(inter)|hole from carpet|color|99.1|98.4|
> |Ours(inter)|hole from carpet|cut|99.8|99.2|
>
> _**Q5. Evaluation of zero-shot performance on BraTS.**_
> > **A5.** Both zero-shot methods [r1] and GAD [52, 61, Ours] train models on one domain (e.g., MVTecAD) and test on another (e.g., VisA or BraTS) to achieve cross-domain generalization. Thus, when BraTS is used as the test set, the training set is MVTecAD, and all classes in BraTS are unseen. Unlike zero-shot methods, which directly test samples without needing reference samples, GAD requires reference samples during testing. For comparison, we report the zero-shot performance of AnomalyCLIP[r1] on BraTS, as shown in the **Table R3**. The results indicate that incorporating references (e.g., ResAD and our methods) improves detection performance, demonstrating the generalization capability of these methods.
> >
> **Table R3. Comparison with zero-shot methods  on BraTS**
>
> |Method (reference number)|I-AUROC|P-AUROC|avg.|
> |:-:|:-:|:-:|:-:|
> |AnomalyCLIP (None)|69.3|93.2|81.3|
> |ResAD ($N^1$)|73.5|91.0|82.3|
> |Ours ($N^1+A^1$)|78.1|96.5|87.3|
>
> ***Q6. AFL's behavior in cluttered scenes and potential for overfitting.***
> > **A6.** According to [1], the MVTecAD dataset comprises both texture and object images. To evaluate the detection capabilities in complex texture scenarios, we specifically analyzed the texture image products from the MVTecAD dataset (carpet, grid, leather, tile, wood). The results are presented in **Table R4**. (Detailed subset-level results are provided in Appendix G for reference.) For comparative purposes, we also included the product-wise performance of PromptAD [25] (drawing data from Tables 11 and 12 of the original paper). The comparison indicates that our method does not exhibit a disadvantage in detecting texture images, even though PromptAD requires training a specific model for each product.
>
> **Table R4. Performance Comparison of Texture Images**
>
> |Products|Ours (I-AUROC)|Ours (P-AUROC)|PromptAD (I-AUROC)|PromptAD (P-AUROC)|
> |:-:|:-:|:-:|:-:|:-:|
> |carpet|100.0|99.3|100.0|95.9|
> |grid|99.8|99.4|99.8|95.1|
> |leather|100.0|99.1|100.0|93.3|
> |tile|99.8|96.1|100.0|92.8|
> |wood|99.8|96.0|97.9|89.4|
>
> > Additionally, while residual features offer some cross-domain capability, the risk of overfitting remains. We observed this phenomenon during training, where training for more epochs negatively impacted model performance to some extent. We mitigated the impact of overfitting by setting fewer epochs (This technique is also used by AnomalyCLIP[r1]) and minimizing the number of training parameters, as described in Appendix F. Since figures cannot be provided here, we will supplement the performance curves regarding training epochs in the main text.
>
> *[r1] AnomalyCLIP: Object-agnostic Prompt Learning for Zero-shot Anomaly Detection. ICLR 2024*

---

> > ### Comment · Reviewer_LrM6 · 2025-08-06
> >
> > Thanks to the authors for their response that addressed my concerns. Browsing through other reviewers' discussions, the point about unfair comparisons also raises new concerns for me. Further Clarification on this point is critical. I will maintain my original score for now.

---

> ### Author Response · Authors · 2025-08-07
>
> Thank you for your feedback. We discuss your concerns in our response regarding **RxZT** and are available to provide further clarification or answer any additional concerns.

---

> > ### Comment · Reviewer_LrM6 · 2025-08-07
> >
> > Thanks for your clarification. My concerns have been addressed. As the response covers many aspects that need to be improved, please include them in the final version. I will keep my score as "borderline accept".

---

> ### Author Response · Authors · 2025-08-07
>
> Thank you for your valuable feedback and positive rating. We will carefully incorporate the comments into the final version.

---

### Note · Authors · 2025-08-13

Dear Reviewers, Area Chairs, and Program Chairs,

We sincerely thank all reviewers for their constructive feedback.

Through this discussion, we believe we have fully addressed each reviewer's concerns, as evidenced by their positive scores. Furthermore, the reviewers' suggestions will be carefully incorporated into the revised paper, which primarily involves some sentence modifications and the incorporation of a few experimental results.

Once again, thank you to all the reviewers for their valuable feedback and positive scores.

---

### Decision · Program_Chairs · 2025-09-17

**Decision:**

Accept (poster)

**Comment:**

This paper introduces the Normal-Abnormal Generalist Learning framework for anomaly detection. In this setting models must generalize across domains using both normal and limited abnormal references. The method comprises Residual Mining, which extracts transferable residual representations and Anomaly Feature Learning, which adaptively identifies anomaly features in query images. Experiments on MVTecAD, VisA, and BraTS demonstrate notable improvements over prior generalist anomaly detection approaches. The paper addresses an important and practical gap in anomaly detection by explicitly incorporating abnormal references. The design is both conceptually sound and empirically validated through extensive experiments. The main weakness is unfair comparisons with existing works (they do not use abnormal samples) and some claims, e.g. first work utilizing abnormal samples as reference, are obivously exggerated. After discussion among reviewers and AC, all reviewers are leaning towards acceptance. Therefore, I recommend to accept this submission and the authors are strongly encouraged to address the remaining issues, in particular, comparison with existing works in the camera ready version.